# Notch2 controls non-autonomous Wnt-signalling in chronic lymphocytic leukaemia

Maurizio Mangolini[1], Frederik Götte[2], Andrew Moore [1], Tim Ammon[2], Madlen Oelsner[2],
Gloria Lutzny-Geier [3], Ludger Klein-Hitpass[4], James C. Williamson[5], Paul J. Lehner[5], Jan Dürig[6],
Michael Möllmann[6], Lívia Rásó-Barnett[7], Katherine Hughes [8], Antonella Santoro[1], Simón Méndez-Ferrer [1,9],
Robert A.J. Oostendorp [2], Ursula Zimber-Strobl[10], Christian Peschel[2,11], Daniel J. Hodson[1],
Marc Schmidt-Supprian[2,11] & Ingo Ringshausen [1]

The Wnt signalling pathway, one of the core de-regulated pathways in chronic lymphocytic leukaemia (CLL), is activated in only a subset of patients through somatic mutations. Here we describe alternative, microenvironment-dependent mechanisms of Wnt activation in malignant B cells. We show that tumour cells specifically induce Notch2 activity in mesenchymal stromal cells (MSCs) required for the transcription of the complement factor C1q. MSC-derived C1q in turn inhibits Gsk3-β mediated degradation of β-catenin in CLL cells. Additionally, stromal Notch2 activity regulates N-cadherin expression in CLL cells, which interacts with and further stabilises β-catenin. Together, these stroma Notch2-dependent mechanisms induce strong activation of canonical Wnt signalling in CLL cells. Pharmacological inhibition of the Wnt pathway impairs microenvironment-mediated survival of tumour cells. Similarly, inhibition of Notch signalling diminishes survival of stroma-protected CLL cells in vitro and disease engraftment in vivo. Notch2 activation in the microenvironment is a pre-requisite for the activation of canonical Wnt signalling in tumour cells.

[1] Wellcome Trust/ MRC Cambridge Stem Cell Institute & Department of Haematology, University of Cambridge, Cambridge CB2 0AH, UK. [2] Department of Hematology and Medical Oncology, Klinikum rechts der Isar der Technischen Universität München, Munich 81675, Germany. [3] Department of Internal Medicine 5, Haematology and Oncology, Friedrich-Alexander-University Erlangen-Nürnberg, Erlangen 91054, Germany. [4] Institute of Cell Biology, Faculty of Medicine, University of Duisburg-Essen, Essen 45122, Germany. [5] Cambridge Institute for Medical Research (CIMR), University of Cambridge, Cambridge CB2 0XY, UK. [6] Department of Hematology, University Hospital Essen,, University of Duisburg-Essen, Essen 45122, Germany. [7] Haematopathology and Oncology Diagnostic Service (HODS), Cambridge University Hospitals NHS Foundation Trust, Cambridge CB2 0QQ, UK. [8] Department of Veterinary Medicine, University of Cambridge, Cambridge CB3 0ES, UK. [9] NHS Blood and Transplant, Cambridge CB2 0PT, UK. [10] Helmholz Zentrum, Research Unit Gene Vectors, Munich 81377, Germany. [11] German Cancer Consortium, DKFZ, Heidelberg 69120, Germany. These authors contributed equally: Maurizio Mangolini, Frederik Götte. Correspondence and requests for materials should be addressed to I.R. (email: ir279@cam.ac.uk)

In recent years, the landscape of genomic mutations in chronic lymphocytic leukaemia (CLL) has become significantly more complex, now allowing the identification of very small sub-clones[1–3]. These mutations cluster in key cellular pathways regulating response to DNA damage, inflammation, chromatin remodelling and RNA processing[4]. In addition, somatic mutations affecting the highly conserved Wnt signalling pathway were reported to be present in a subset of CLL patients[5]. The expression of the key transcriptional factor β-catenin is tightly regulated through post-translational mechanisms. Functional in vitro studies demonstrated that the survival of leukaemic B cells is indeed dependent on the Wnt pathway, particularly in patients carrying these mutations[5,6]. In vivo evidence for the significance of the Wnt pathway originate from studies in the Tcl1 model, demonstrating that the deletion of the Wnt receptor Fzd6 significantly delays tumourigenesis[7].

In spite of the success in deciphering the genomic complexity of CLL, no universal drivers of the disease have been identified. Instead, the emerging picture of the pathogenesis of CLL is that these mutations operate in conjunction with tumourigenic cues from the microenvironment[8,9]. In this sense, malignant B cells do not survive nor proliferate autonomously, but these aspects of lymphomagenesis are co-dependent on bystander cells of the microenvironment. Among these, bone marrow-derived mesenchymal stromal cells (BMSCs) play a central role in providing pro-survival factors for CLL cells. Work from many groups demonstrated that BMSCs regulate various biological processes in leukaemic B cells, including metabolic changes[10,11], alterations in the expression of surface[12,13] and anti-apoptotic proteins[14], thereby contributing to drug resistance[15]. Notably, the communication between CLL cells and BMSCs is bi-directional, also resulting in morphological and transcriptional changes in stromal cells[16]. Here we describe a signalling pathway underlying the mutual activation of BMSCs and tumour cells, which depends on the CLL-mediated activation of Notch2 in stromal cells, and the reciprocal activation of the canonical Wnt pathway in CLL cells.

## Results

**MSCs induce gene expression reprogramming in CLL cells.** CLL is characterised by an enormously diverse spectrum of disease-associated mutations. Correspondingly, the fitness of cells cannot solely be maintained by cell-intrinsic signals, but is exquisitely dependent on cues provided by the tumour microenvironment. We previously established a co-culture system to investigate the heterotypic interactions between stromal cells and primary CLL cells[16]. EL08-1D2 cells are primary stromal cells derived from mouse embryonic (E11) livers supporting human haematopoietic stem cell (HSC) activity[17]. To assess to what extent mesenchymal stromal cells can influence the biology of CLL cells, we performed deep RNA sequencing on purified CLL cells, obtained from 6 individual untreated patients, cultured on EL08-1D2 cells for 48 h. Stringent filtering was applied by considering only those significant genes (uncorrected $p < 0.05$), which showed raw reads >20 in ≥ 4 samples. Comparison of these results to RNA-sequencing (RNA-seq) data from the respective mono-culture in the absence of stromal cells showed an up-regulation of 2268 genes (>2-fold) and a down-regulation of 1076 genes (>2-fold; Fig. 1a). Subjecting this transcriptomic data to Gene Set Enrichment Analysis (GSEA) identified transcriptional changes in gene sets involved in cell adhesion, cell metabolism and signalling (Fig. 1b and Supplementary Figure 1a,b and 2a). Figure 1c depicts a heat map of the 50 most significantly up- and down-regulated genes. Notably, among those induced, genes affecting remodelling of the microenvironment are highly over-represented. To investigate which signalling pathways in CLL

cells mediate these transcriptional changes, we performed a multiplex immunoblot assay from CLL cells, co-cultured for 6 h on EL08-1D2 cells or mono-cultured. Results from this experiment indicate that BAD, PRAS40 and glycogen synthase kinase 3-β (GSK3-β) were phosphorylated by direct contact to stromal cells (Fig. 1d; to facilitate reading, throughout the figures, proteins detected in B cells are labelled in blue, stromal cell-derived proteins in black). As these proteins are known targets of AKT, our findings are in line with data suggesting that contact to stromal cells triggers a pro-survival signal via phosphatidylinositol-3-kinase and AKT in CLL cells[18]. In contrast to HSCs[19], the provision of pro-survival effects of EL08-1D2 cells on primary CLL cells requires a direct cell–cell contact, as disruption of this direct cell interaction abolishes anti-apoptotic signals to CLL cells (Fig. 1e and Supplementary Figure 2b, Supplementary Figure 11a shows the gating strategy used to detect live and apoptotic CLL cells). This demonstrates that some of the most important signals from the stroma to CLL cells cannot be transmitted by soluble factors but rely on direct cellular interactions.

**CLL cells activate Notch2 signalling in BMSCs.** Notably, as shown previously[20,16], CLL cells are able to receive anti-apoptotic signals from numerous different mouse and human mesenchymal cells, suggesting that this cell–cell communication is based on evolutionary highly conserved signalling principles. Since the Notch pathway constitutes such an ancient signalling pathway, we investigated whether the crosstalk between BMSCs and CLL cells involves Notch signalling. Firstly, we assessed the expression of Notch ligands on primary CLL cells. In agreement with a previous report[21], CLL cells constitutively express the Notch ligands Jagged-1, Jagged-2 and Delta (DLL) (Fig. 2a and Supplementary Figure 3a). On the opposing EL08-1D2 cells, Notch1 and Notch2 had by far the highest surface expression (Fig. 2b). This expression profile was very similar in cultured primary mouse BMSCs (Supplementary Figure 3b). To exclude the possibility that Notch expression was related to in vitro propagation of BMSCs, we investigated Notch expression in BMSCs from 10-week-old Nestin[GFP] mice, which express green fluorescent protein (GFP) in a subset of BMSCs with colony-forming unit fibroblastic capacity[22]. We observed Notch1 and Notch2 expression in Nestin[GFP]-MSCs, endothelial cells and osteoblasts, although expression levels were significantly lower than on in vitro cultured BMSCs (Supplementary Figure 3c; Supplementary Figure 11b shows the gating strategy used to detect various BMSC populations). The presence of Notch ligands on CLL cells and receptors on BMSCs suggested that CLL could activate Notch signalling in stromal cells. Since Notch3 and Notch4 lack a classical transactivation domain and were only expressed at comparatively low levels on EL08-1D2 cells, we focussed on the function of Notch1 and Notch2. Upon binding of Notch ligands, two consecutive proteolytic cleavages release the Notch intracellular domain (ICD), which then acts as a transcription factor by interacting with the DNA binding protein CSL. Surprisingly, although Notch1 was expressed on EL08-1D2 cells in mono-culture, cleaved Notch1 levels significantly decreased after contact to primary CLL cells (Fig. 2c). In contrast, cleaved Notch2 levels increased in EL08-1D2 cells and primary human BMSCs following activation by CLL cells (Fig. 2d, e and Supplementary Figures 4a,b). In line with this observation, nuclear Notch2 levels became more abundant in BMSCs co-cultured with CLL cells (Fig. 2f). The up-regulation of total Notch2 in BMSCs was abrogated by the presence of a gamma-secretase inhibitor (Fig. 2g). No activation of Notch3 or Notch4 was detected in CLL-activated stromal cells (Supplementary Figure 4c). These data indicate that malignant B cells specifically activate Notch2 signalling in BMSCs.

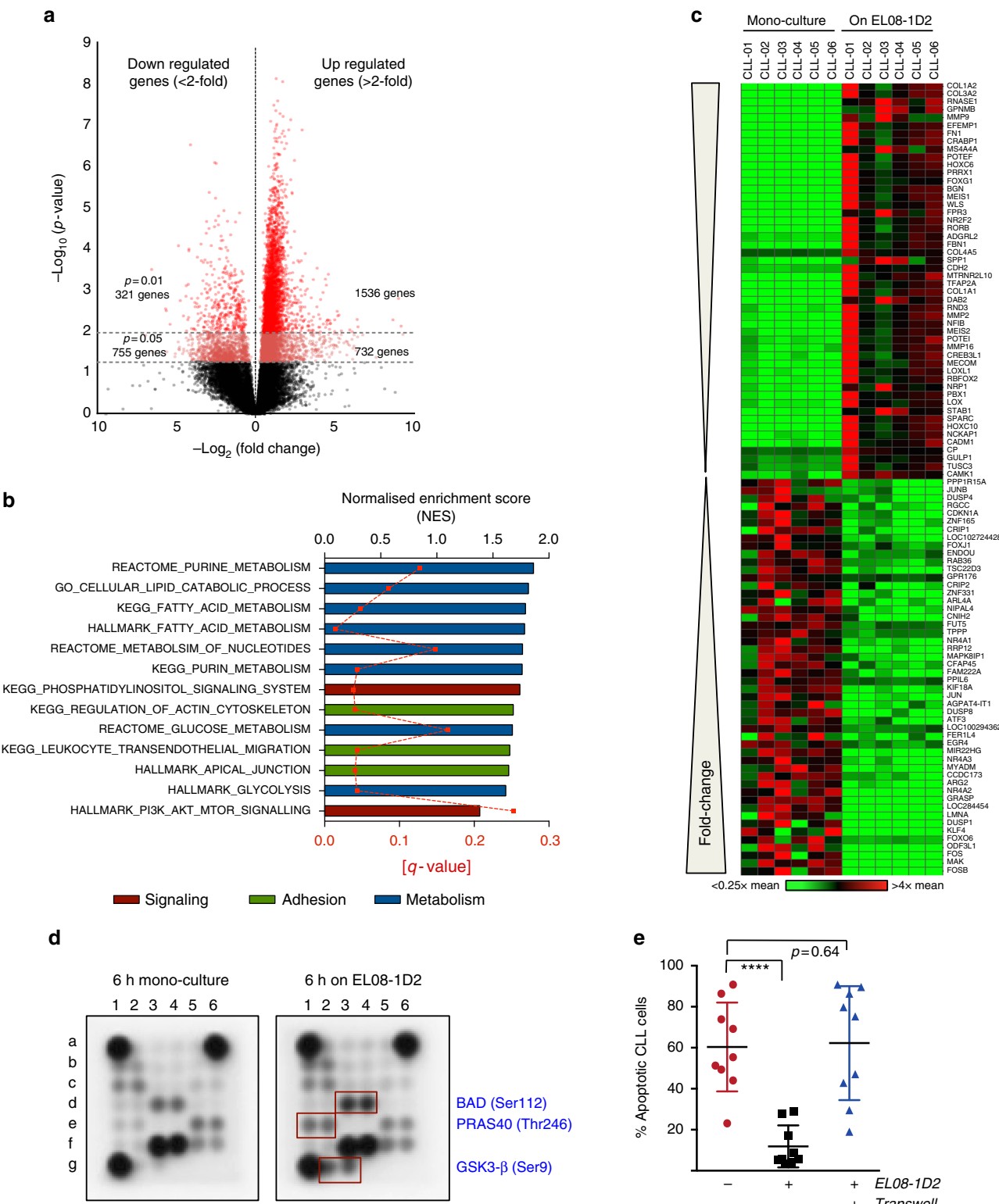

**Fig. 1** Activation of CLL cells by BSMCs. **a** Volcano plot showing the differentially up- and down-regulated genes in CLL cells after 48 h of co-culture on EL08-1D2 cells compared to cells cultured for 4 h in mono-culture (4 h was chosen to avoid gene expression changes related to cell death). RNA-sequencing was performed on samples from 6 individual patients. **b** Transcriptomic data were subjected to Gene Set Enrichment (GSEA) analyses to identify pathways in CLL activated by contact to stromal cells. Gene sets are listed in order of Normalised Enrichment Scores (top black X-axis). FDR q values for each gene set are indicated by the red dotted line (lower red X-axis). **c** Heat map showing the 50 most significantly up- and down-regulated genes in CLL cells in response to contact with stromal cells. **d** Cell extracts from CLL mono-culture or from cells cultured for 6 h on EL08-1D2 cells were analysed using a human intracellular phosphorylation antibody array. Representative results from four different patients and experiments are shown. **e** CLL cells were cultured in medium only (red circles) or on EL08-1D2 cells (black squares) for 5 days before analysing apoptotic cells by Annexin-V/PI staining. Transwells were used to disrupt direct cell–cell contacts (blue triangles). Error bars show mean ± SEM from 9 patients; ****$p < 0.0001$

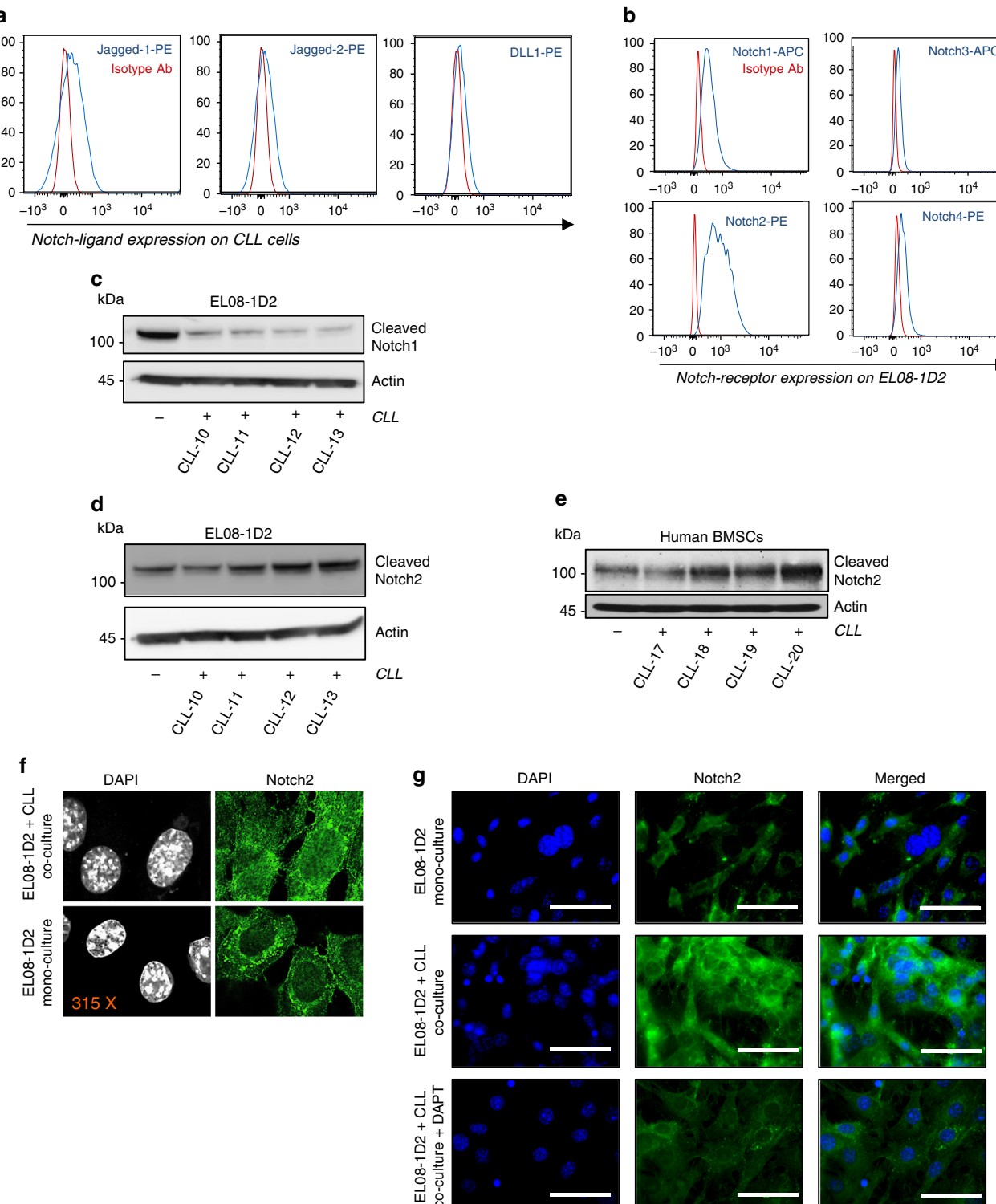

**Fig. 2** CLL cells activate Notch2 in BMSCs. **a** Constitutive expression of the Notch ligands Delta-1, Jagged-1 and Jagged-2 was analysed in primary CLL cells by flow cytometry. Representative results from three different patients are shown. **b** Constitutive expression of Notch1-4 in primary EL08-1D2 cells was assessed by flow cytometry. Two separate experiments revealed identical results. **c** Notch1 activity in EL08-1D2 cells was assessed by immunoblotting for Notch1[ICD] 48 h after co-culturing primary CLL cells from 4 different patients on stromal cells. **d** Notch2 activity in EL08-1D2 cells was assessed by immunoblotting for Notch2[ICD] 48 h after co-culturing primary CLL cells from the same 4 patients as shown in (**c**). **e** Notch2[ICD] levels were analysed by western blotting in human bone marrow stromal-derived cells and human bone marrow stromal-derived cells co-cultured with CLL primary cells. **f** Confocal microscopy analysis of Notch2 expression in EL08-1D2 following 48 h co-culture with primary CLL cells or mono-culture. Images were captured with identical exposure time and settings between mono-cultures and co-cultures. Representative images from three independent experiments are shown. **g** Notch2 expression on EL08-1D2 cells was assessed on mono-cultured cells or on cells which were co-cultured for 48 h in the absence or presence of 20uM DAPT. Scale bar = 100 μm. One representative experiment out of three is shown

**Notch2 regulates distinct gene sets in activated BMSCs**. To gain insights into the transcriptional program orchestrated by Notch2, we generated Notch2-deficient BMSC from Notch2 conditional knockout (*Notch2^fl/fl*) mice[23] and ablated Notch2 by treatment with Cre protein (HTNC) (Fig. 3a). Successful ablation of Notch2 protein was confirmed by flow cytometry and western blotting (Fig. 3b). Subsequently primary CLL cells were cultured on Notch2-proficient and -deficient confluent monolayers of BMSCs. After 5 days, RNA was isolated from purified stromal cells and gene expression profiles were assessed using the Affymetrix GeneChip platform. Unsupervised analyses of transcriptomes using principle component analysis indicate that Notch2 constitutively regulates genes in BMSCs in the absence of CLL cells (Fig. 3c, compare light blue to orange clusters; enriched genes sets are presented in Supplementary Figure 5a). However, activation of stromal cells by direct contact to malignant B cells induces significant Notch2-dependent changes in their gene expression profiles (Fig. 3c, red and dark blue clusters). Functional enrichment analyses of transcriptomes from CLL co-cultured stromal cells confirmed a Notch gene expression signature present in *Notch2^fl/fl* cells (Fig. 3d and Supplementary Figure 5b). In addition, Notch2 regulates genes involved in inflammation and extracellular matrix formation, both important components of the CLL microenvironment (Fig. 3d). After applying an arbitrary cut-off of twofold and a false discovery rate (FDR) of <0.01, we identified 76 annotated genes induced and 155 repressed (Supplementary data 1 and 2) in BMSCs by Notch2. Only after contact to CLL cells, 88 and 74% of the respective gene sets were regulated (Fig. 3e). The heat map in Fig. 3f depicts the genes that were most strongly up- or down-regulated in Notch2-deficient and CLL-activated BMSCs. We then subjected consistently altered target probe sets to overrepresentation analysis using the Gene-Trail software and found significant enrichments for the gene ontology (GO) terms *regulation of cell adhesion* ($p = 0.0007$) and *integrin-binding* ($p = 0.002$). Notably, these analyses also identified *Wnt-protein binding* ($p = 0.029$) as a gene cluster regulated by Notch2. Genes with known functions in these pathways or in CLL biology are listed in order of their degree of regulation by Notch2 (Fig. 3f, right panel).

**Notch2 in BMSCs is required for Wnt signalling in CLL cells**. Results from the transcriptome analyses indicated that stromal Notch2 can impinge on canonical Wnt signalling. To investigate whether Wnt signalling in CLL cells is affected by contact to BMSCs, we analysed primary, peripheral blood-derived CLL cells for the expression of β-catenin. The constitutive expression of β-catenin in CLL cells is extremely low, consistent with previous reports[24,25]. Co-culture of malignant B cells with primary mouse or human BMSCs and with EL08-1D2 cells strongly induced β-catenin expression (Fig. 4a–c). Expression gradually increased with time and was contact-dependent, as conditioned media from EL08-1D2/ CLL co-cultures failed to stabilise β-catenin in the absence of stromal cells (Fig. 4c). However, western blotting revealed several weaker bands with a higher molecular weight in CLL cells cultured in conditioned media, suggestive of partly stabilised but ubiquitinated β-catenin (Fig. 4c). Stabilised β-catenin accumulates in the cytoplasm and subsequently travels to the nucleus where it regulates gene expression as a co-factor to the T-cell factor/ lymphoid enhancer factor (TCF/LEF). To demonstrate nuclear expression of β-catenin in BMSC-activated CLL cells, we performed immunofluorescence with antibodies recognising either activated or total β-catenin. Our results showed that activated, de-phosphorylated β-catenin is predominantly localised in the nucleus (Fig. 4d). Cellular fractionation and immunoblotting of previously co-cultured CLL cells confirmed

these findings (Supplementary Figure 6a). Comparison of β-catenin levels in primary CLL cells cultured on Notch2-proficient or -deficient BMSCs indicated that Notch2 expression in BMSCs was indeed an essential requirement for β-catenin stabilisation (Fig. 4e). To confirm this result by different means we engineered Notch2-deficient EL08-1D2 cells using CRISPR/Cas9 (Clustered Regularly Interspaced Short Palindromic Repeats/CRISPR associated protein 9)-mediated deletion (Supplementary Figure 6b). Importantly, Notch2 ablation did not affect BMSCs or EL08-1D2 cell morphology or proliferation (Supplementary Figure 6c,d). Loss of Notch2 in EL08-1D2 cells impaired β-catenin accumulation in co-cultured CLL cells and essentially abolished nuclear β-catenin (Fig. 4f). Quantification of β-catenin levels in 20 individual, co-cultured CLL samples indicated various levels of dependency on stromal Notch2 (Supplementary Figure 6e; patient characteristics are provided in Supplementary Table 1), which possibly reflects the contribution of CLL-intrinsic mutations to the activation of the Wnt pathway. Since GSK3-β induces the degradation of β-catenin, we investigated whether stromal Notch2 affected the activity of GSK3-β in CLL cells. Contact of primary B cells to EL08-1D2 cells inhibited GSK3-β, indicated by increased inhibitory phosphorylation on serine 9 (Figs. 1d, 4g). Loss of stromal Notch2 essentially prevented the inhibition of GSK3-β by stroma contact. Furthermore, chemical inhibition of GSK3-β with lithium chloride restored β-catenin levels in CLL cells cultured on Notch2-deficient BMSCs (Fig. 4h). These data demonstrate that Notch2, activated in BMSCs by malignant B cells, is required to reciprocally activate canonical Wnt signalling in CLL cells through the inhibition of GSK3-β.

Our transcriptome analysis indicated that Notch2 controls the expression of the complement factor C1q in BMSCs. As previously shown, C1q can activate the Wnt co-receptor LRP5/6, thereby inducing β-catenin stabilisation[26]. To assess whether stromal C1q mediates Notch2-dependent activation of β-catenin, we first analysed C1q RNA expression in EL08-1D2 cells. Similar to our results from conditional Notch2-deficient murine BMSCs (Fig. 3f and Supplementary Data 1+2), CRIPSR/Cas9 deletion of Notch2 also down-regulated complement C1q messenger RNA (mRNA) expression in CLL-activated EL08-1D2 cells (Fig. 4i). To assess whether complement C1q is required for the GSK3-β-mediated stabilisation of β-catenin in CLL cells, we supplemented C1q to CLL-EL08-1D2 co-cultures. Addition of C1q compensated for Notch2 deficiency of stromal cells and inhibited GSK3-β-mediated degradation of β-catenin in CLL cells (Fig. 4j). In contrast, C1q had no effect on CLL mono-cultures in the absence of stromal cells (Supplementary Figure 6f). These data indicate that stromal Notch2 controls complement C1q production, which inhibits GSK3-β and stabilises β-catenin in malignant B cells.

**Stromal N-cadherin mediates stabilisation of β-catenin**. Stabilisation of β-catenin in the cytoplasm is a key mechanism of regulating its transcriptional activity. In addition to proteasomal destruction of β-catenin in the absence of Wnt activators, β-catenin levels are also regulated by binding to the cytoplasmic domain of class-I cadherins, which mediate cell–cell adhesion through homotypic protein interactions. Since N-cadherin was shown to be crucial for the homing of normal and malignant HSC to mesenchymal cells[27], we analysed the expression of N-cadherin in CLL cells. Peripheral blood-derived CLL cells lacked N-cadherin expression, which was progressively up-regulated upon culture on EL08-1D2 cells (Fig. 5a) and human stromal cells (Supplementary Figure 7a). Importantly, the kinetic of N-cadherin induction in CLL cells mirrored that of β-catenin, suggesting a functional connection. Co-immunoprecipitation of N-cadherin or β-catenin demonstrated their direct interaction in

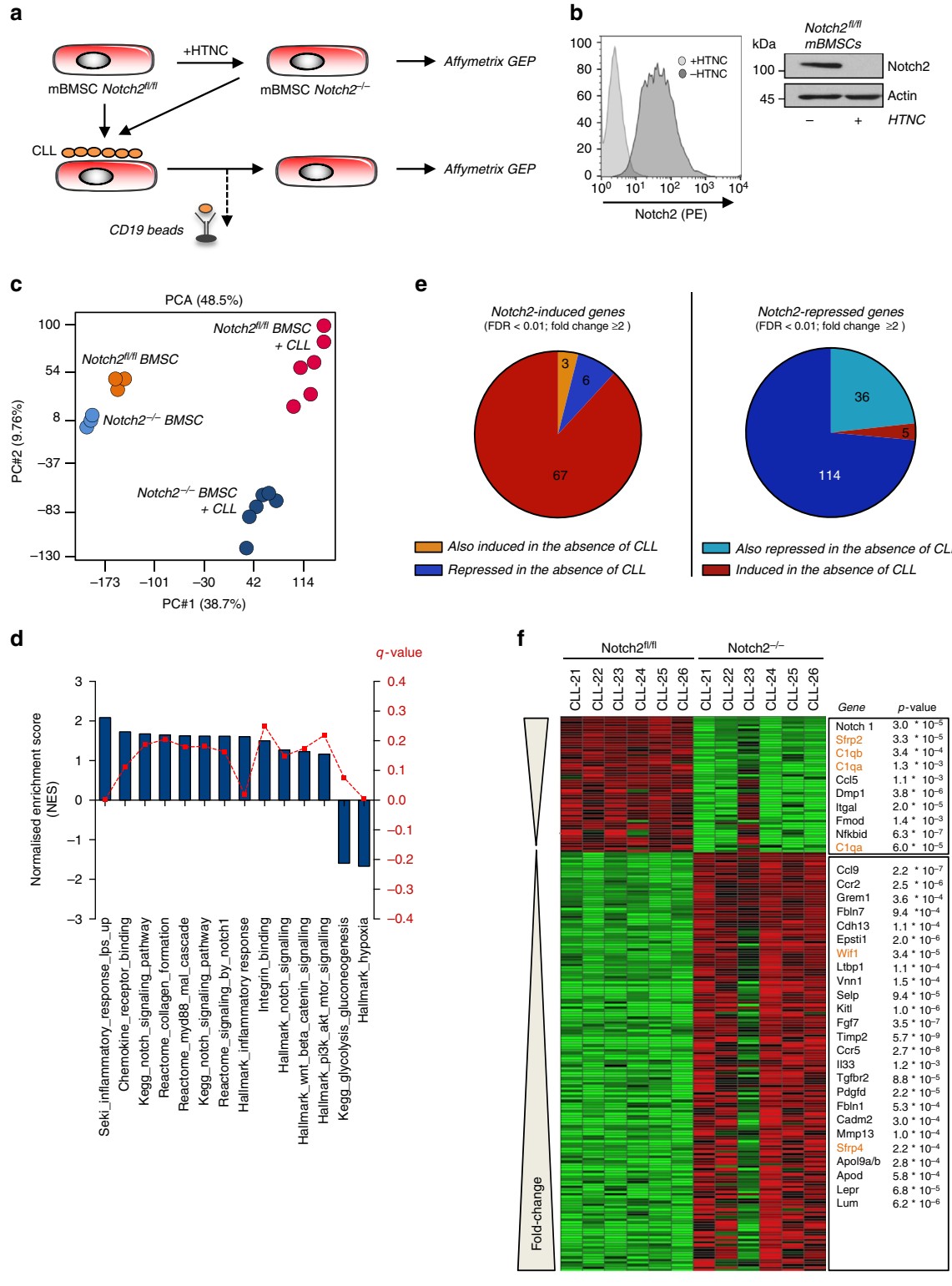

**Fig. 3** Notch2 gene regulation in BMSCs. **a** Schematic representation of the experimental model. **b** Flow cytometry and immunoblot analyses of Notch2 in *Notch^fl/fl^* stromal cells following in vitro CRE recombination. **c** Principal component analyses of the transcriptomes from three Notch2-proficient (orange circles) and Notch2-deficient (light blue circles) BMSCs and from five Notch2 wild-type (red circles) or knockout (dark blue) BMSCs cultured with CLL cells for 5 days. **d** GSEA comparing the expression of genes associated with the presence or absence of Notch2 expression in BMSCs co-cultured with primary CLL cells. Gene sets are listed in order of Normalised Enrichment Scores (left black *Y*-axis). FDR *q* values for each gene set are indicated by the red dotted line (right red *Y*-axis). **e** Pie-chart of Notch2-induced (left) and Notch2-repressed genes (right). **f** Heat map showing the 200 most significantly up- and down-regulated genes in Notch2-deficient BMSCs in response to contact with CLL cells. Genes listed in red have known functions in regulating Wnt signalling

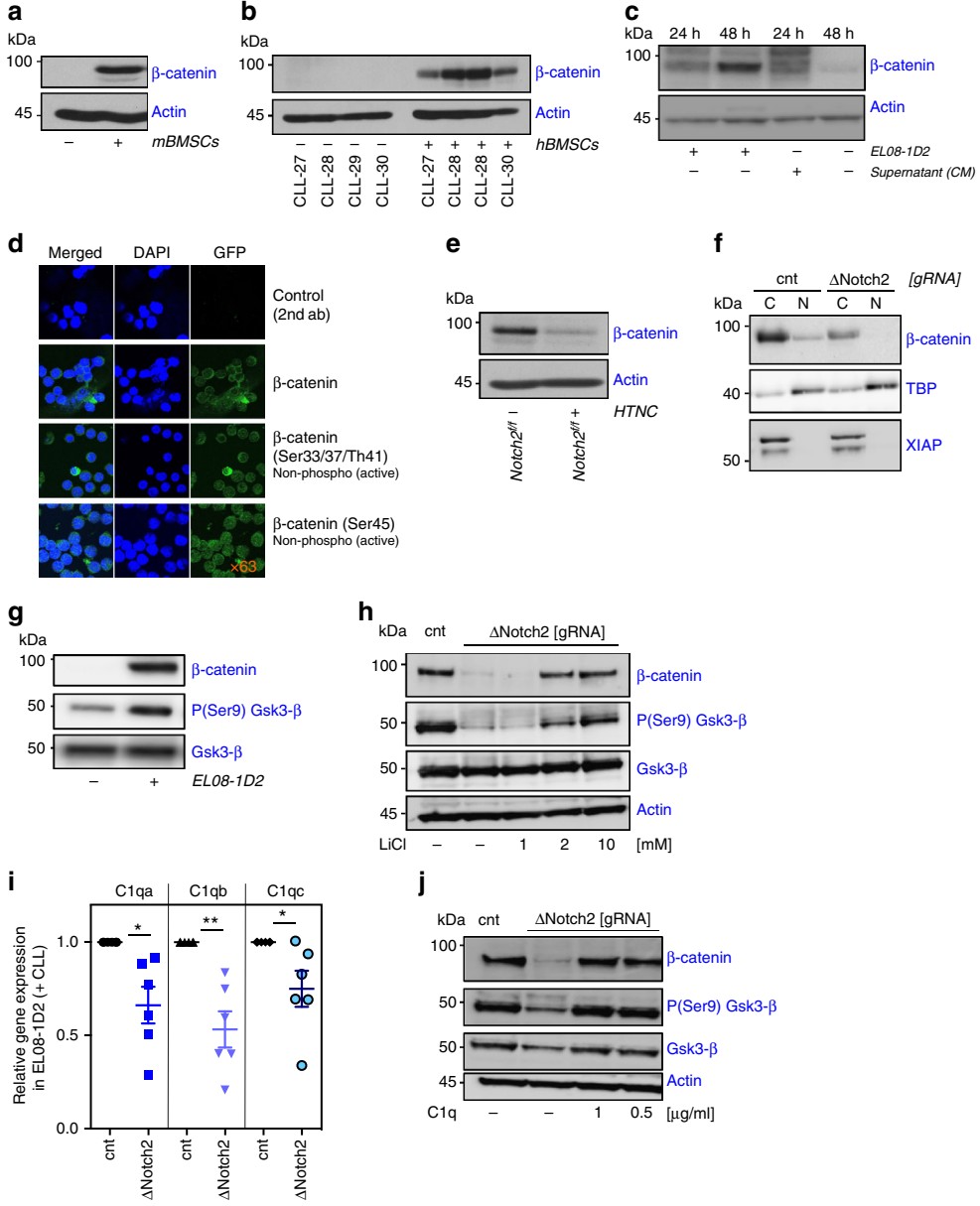

**Fig. 4** Stromal Notch2 regulates β-catenin in CLL cells. **a** β-Catenin expression in primary CLL cells co-cultured on murine BMSCs for 5 days or mono-cultured. **b** β-Catenin expression in four different primary CLL cells co-cultured on human primary BMSCs. **c** Lane 1+2: β-catenin expression was evaluated in primary CLL cells after 24 and 48 h in direct co-culture with EL08-1D2. Lane 3: conditioned media (CM) from the 48 h co-cultures or fresh medium (lane 4) was used as culture medium for freshly thawed cells of the same patient. β-Catenin expression was assessed after 24 h. Representative results from three different patients are shown. **d** β-Catenin localisation was assessed in primary CLL cells co-cultured on EL08-1D2 cells for 24 h. Specific non-phospho antibodies were used to detect the active form of β-catenin. Representative results from two different patients are shown. **e** β-Catenin expression in primary CLL cells cultured for 5 days on Notch2$^{+/+}$ or Notch2$^{-/-}$ mBMSCs. **f** Analysis of cytoplasmic and nuclear β-catenin levels in primary CLL cells co-cultured for 48 h on BMSCs in which Notch2 was deleted by CRISPR/Cas9. Representative results from three different patients are shown. **g** Phospho-GSK3-β expression in primary CLL cells co-cultured on EL08-1D2 cells for 24 h. Representative results from three different experiments are shown. **h** β-Catenin expression in CLL cells co-cultured on Notch2-deficient EL08-1D2 cells. After 24 h on stromal cells, co-cultures were exposed to increasing doses of lithium chloride (LiCl) for additional 3 h before CLL cells were harvested. One representative experiment out of three is shown. **i** Quantitative reverse-transcription polymerase chain reaction analysis of the C1q complex in Notch2-deficient EL08-1D2 (blue symbols) cells normalised to expression in control cells (transfected with a control guide RNA, dark symbols). Shown is the mean ± SEM of six independent experiments, using different primary CLL cells; *$p < 0.05$, **$p < 0.01$. **j** β-Catenin expression in CLL cells co-cultured on Notch2-deficient EL08-1D2 cells. After 24 h on stromal cells, co-cultures were exposed to increasing doses of C1q for additional 24 h before CLL cells were harvested. Representative results from three different experiments are shown

CLL cells (Fig. 5b and Supplementary Figure 7b). In contrast to β-catenin, N-cadherin levels were transcriptionally regulated in CLL cells (Fig. 5c). In order to address whether homotypic N-cadherin interactions between CLL cells and BMSCs were required to stabilise β-catenin, we generated N-cadherin-deficient stromal cells. Depletion of N-cadherin from EL08-1D2 cells did not affect their baseline expression of Notch2 (Fig. 5d). Of note, loss of N-cadherin also did not affect expression levels of E-cadherin (Supplementary Figure 7c). However, N-cadherin-deficient stromal cells failed to induce N-cadherin expression in primary CLL

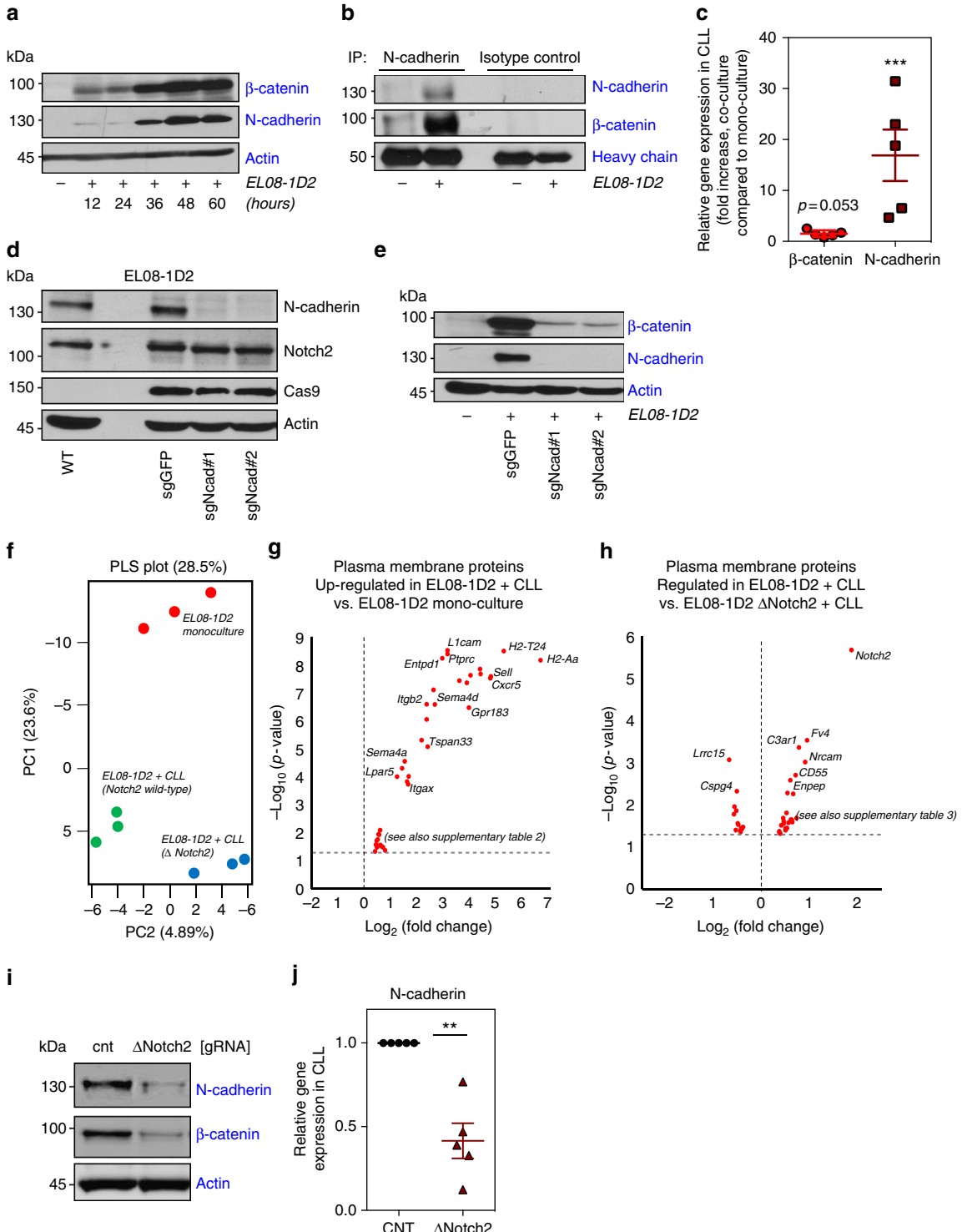

cells (Fig. 5e). In agreement with the idea that N-cadherin contributes to the stabilisation of β-catenin, this lack of N-cadherin induction in CLL cells was associated with significantly reduced β-catenin levels (Fig. 5e). Thus, our data demonstrate that in addition to Notch2-mediated GSK3-β inhibition in CLL cells, full stabilisation of β-catenin in malignant B cells also requires N-cadherin expression induced by stromal N-cadherin.

Since conditioned media from CLL/BMSC co-culture failed to fully stabilise β-catenin in CLL cells (Fig. 4c), membrane-bound factors must also be required to inhibit its degradation. We therefore performed plasma membrane profiling of EL08-1D2

cells, which allows the quantitative proteomic analysis of cell surface proteins[28] in the presence and absence of Notch2. Plasma membrane proteins were labelled as described in the Methods section *after* CLL cells were removed from stromal cells. Only stromal cells with a purity of >98% were further analysed by mass spectrometry (Supplementary Figure 7d). Following mass spectrometry analysis, we identified 1055 plasma membrane proteins as defined by GO terms, representing 34% of the total number of proteins identified. This subset accounted for 64% of total abundance of proteins identified by mass spectrometry, indicating successful enrichment for plasma membrane proteins

**Fig. 5** Stabilisation of β-catenin is partially dependent on N-cadherin. **a** CLL cells were co-cultured on EL08-1D2 cells. After the time points as indicated, N-cadherin and β-catenin expression were analysed. Two additional experiments revealed similar results. **b** N-cadherin was immunoprecipitated from CLL lysates derived from mono-cultures or from co-cultures with EL08-1D2 cells. One out of three experiments is shown. **c** Quantitative reverse-transcription polymerase chain reaction analysis of β-catenin (red circles) and N-cadherin (ruby squares) mRNA expression in primary CLL cells 24 h after co-culture with EL08-1D2, normalised to expression in mono-cultured cells. Shown is the mean ± standard deviation of five independent primary CLL cells; ***$p <$ 0.001. **d** Cas9-expressing EL08-1D2 cells were transfected with two different guide RNAs (sgRNAs) against N-cadherin. Constitutive expression of N-cadherin and Notch2 is shown. **e** CLL cells were co-cultured on N-cadherin-proficient or -deficient EL08-1D2 cells. Expression of β-catenin and N-cadherin in CLL cells is shown after 72 h. One representative experiments out of three is depicted. **f** Partial least square (PLS) analysis of all proteins detected and quantified by mass spectrometry. Three biological replicates were performed with one CLL patient sample. **g** Volcano plot showing the differentially up- and down-regulated plasma proteins in EL08-1D2 cells after 48 h of co-culture with CLL cells compared to EL08-1D2 cells in mono-culture. **h** Volcano plot showing the differentially up- and down-regulated plasma proteins in EL08-1D2 cells after 48 h of co-culture with CLL cells. Comparison between Notch2-proficient and Notch2-deficient stromal cells. **i** Analysis of N-cadherin and β-catenin levels in primary CLL cells co-cultured for 48 h on BMSCs in which Notch2 was deleted by Cas9. Representative results from three different patients are shown. **j** Quantitative reverse-transcription polymerase chain reaction analysis of N-cadherin mRNA expression in primary CLL cells 48 h after co-culture on Notch2-deficient EL08-1D2 cells (ruby triangles), normalised to expression in control cells (black circles). Shown is the mean ± standard deviation of five independent experiments with individual primary CLL cells; **$p <$ 0.01

(Supplementary Figure 7e). Partial least square analysis (PLS) of three biological replicates showed a clear difference in clustering of proteins between EL08-1D2 cells in mono-culture and in co-culture with CLL cells in the presence or absence of Notch2 (Fig. 5f). After applying a cut-off of twofold change and a p value of 0.05, 25 proteins were differentially expressed between EL08-1D2 cells in mono-culture compared to co-culture with CLL cells (Fig. 5g). We identified plasma membrane proteins induced by CLL cells in stromal cells with known functions in cell adhesion and cell contact-induced signalling (Supplementary Table 2). Comparison of results from Notch2-proficient to -deficient stromal cells showed that the experiment was successful as Notch2 was identified as the most differentially expressed cell surface protein (Fig. 5h). We found an additional 36 proteins regulated by Notch2 expression in stromal cells with a p value of <0.05, but with a less than twofold difference (Supplementary Table 3).

Notably, several of the Notch2-regulated proteins are growth factor receptors, possibly contributing to Notch2-dependent activation of CLL cells. However, N-cadherin expression on stromal cells was itself not affected by loss of Notch2 (Fig. 5h and Supplementary Figure 7f).

We next analysed whether the induced expression of N-cadherin in CLL cells was affected by Notch2 activation in stromal cells. Deletion of stromal Notch2 significantly reduced the expression of N-cadherin in CLL cells, accompanied by reduced β-catenin levels (Fig. 5i). Quantification of N-cadherin mRNA indicated that the reduced expression was due to transcriptional down-regulation of N-cadherin (Fig. 5j). Based on these data, we hypothesised that N-cadherin transcription may be regulated by β-catenin. In support of this hypothesis, we found decreased N-cadherin mRNA levels in stroma-activated CLL cells in the presence of the β-catenin inhibitor ICG-001 (Supplementary Figure 7g). In conclusion, these data demonstrate that homotypic interactions between N-cadherin, expressed on BMSCs and malignant B cells, are required for the full stabilisation of β-catenin in CLL cells and are partially controlled by Notch2 activity in BMSCs.

**Therapeutic targeting of the Notch2/ β-catenin crosstalk.** The activation of β-catenin in CLL cells is implicated in defective apoptosis and contributes to disease progression in the Tcl1-mouse model[5–7]. Since Notch2 activity in BMSCs positively regulates β-catenin levels in malignant B cells, we investigated whether the survival of CLL cells cultured on stromal cells was compromised by the lack of Notch2 in BMSCs. Viability of primary CLL cells was assessed after 60 h. Lack of stromal Notch2

only marginally impaired survival of CLL cells compared to wild-type BMSCs (Fig. 6a). We hypothesise that the residual expression of β-catenin in malignant B cells (Fig. 4e and Supplementary Figure 6e) may be sufficient to protect malignant B cells from spontaneous apoptosis under these conditions. Next, we used small molecule inhibitors of Notch or Wnt signalling (illustrated in Fig. 6b) to investigate their effects on CLL cell survival. Importantly, none of the inhibitors showed direct toxicity on EL08-1D2 cells (Supplementary Figure 8a). However, since Wnt activation has been observed in BMSCs, we cannot fully exclude the contribution of direct effects of these inhibitors to stromal cell-mediated survival of CLL cells. Tumour cells were exposed to different inhibitor doses for 48 h to assess their impact on viability in established co-cultures or in CLL mono-cultures. DAPT [$N$-[$N$-(3,5-difluorophenacetyl)-L-alanyl] is a non-transition state inhibitor of the γ-secretase, required for the final cleavage of truncated Notch to release Notch ICD. Treatment of BMSC/CLL co-cultures with DAPT effectively blocked Notch activity in BMSCs as indicated by a down-regulation of HES-1 (Fig. 6c). This correlated with a dose-dependent pro-apoptotic effect of DAPT in CLL cells cultured on EL08-1D2 cells (Fig. 6d) accompanied by a loss of β-catenin in CLL cells (Fig. 6e). Similarly, Wnt pathway inhibitors affected survival of malignant B cells: XAV939, which stabilises the deconstruction complex member axin, significantly impaired survival of CLL cells in co-culture (Fig. 6f, left panel and Supplementary Figure 8b) but not in mono-culture. Dvl-PDZ3, a peptide inhibitor of Dishevelled proteins, blocks β-catenin signalling via direct interference with the Wnt receptor (Fig. 6f-middle panel). This inhibitor also enhanced apoptosis of CLL cells activated by BMSCs. ICG-001 blocks β-catenin signalling further down-stream by interfering with the recruitment of the transcriptional co-activator CBF (C-repeat binding factor) to the β-catenin/TCF complex. Although we observed that ICG-001 slightly exacerbated cell death in mono-culture, its pro-apoptotic effects were significantly enhanced in activated CLL cells in co-culture (Fig. 6f, right panel). In conclusion, these data indicate that inhibition of β-catenin activation reduces the microenvironment-mediated survival of malignant B cells.

To address whether targeting of Notch signalling can also be therapeutically meaningful in vivo, we transplanted $40 \times 10^6$ splenocytes from diseased $E\mu$-$Tcl1$-$Tg$ mice into syngeneic C57BL/6 mice. After 4 weeks of engraftment, mice were treated daily with DAPT (30 mg/kg body weight) or vehicle control (intraperitoneal (i.p.) injection) for 3 consecutive days (Fig. 7a). Flow cytometry analysis from Sca1$^+$ sorted BMSCs showed a reduction of surface Notch2 expression on stromal cells derived

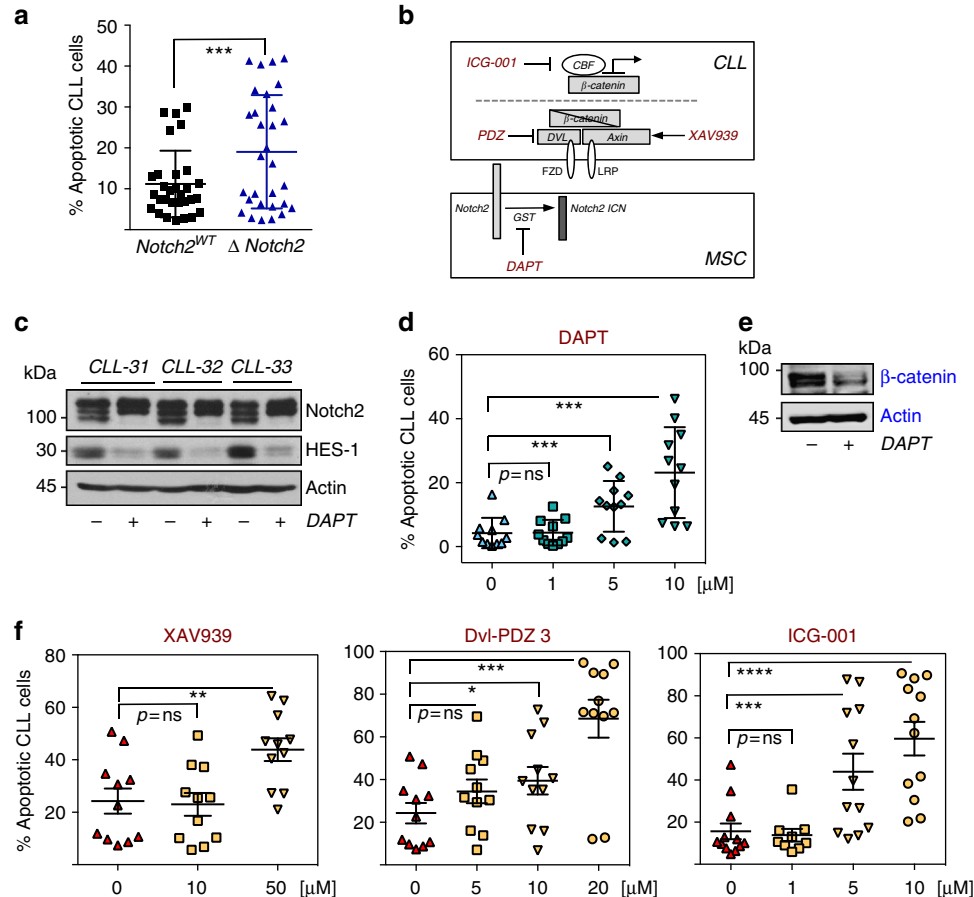

**Fig. 6** Increased sensitivity to Wnt inhibitors in primary CLL cells in the presence of BMSCs. **a** CLL cells were cultured for 5 days on mouse-derived BMSCs, either proficient (black squares) or deficient (blue triangles) for Notch2 (combining results from Cre-mediated and CRISPR/Cas9-mediated deletion of Notch2). Apoptotic cells were analysed by Annexin-V/PI staining. Error bars show mean ± SEM from 30 patients; ***$p = 0.0004$. **b** Schematic representation of the targeted proteins used in this study. **c** Three different CLL cells were co-cultured on EL08-1D2 cells for 72 h in the absence or presence of DAPT. Notch signalling was assessed by immunoblotting EL08-1D2 cell lysates for the expression of HES-1. **d** CLL cells were co-cultured on EL08-1D2 cells for 72 h. Then, increasing doses of DAPT were added for 48 h. Apoptotic CLL cells were detected by flow cytometry and staining for Annexin-V/PI. Error bars show mean ± SEM from 11 patient samples. **e** Analysis of β-catenin levels in primary CLL cells co-cultured for 48 h on BMSCs in the presence of DAPT. A representative result from three different patients is shown. **f** Similarly to data shown in (**d**), CLL co-cultures were exposed to increasing doses of the Wnt inhibitors XAV939, Dvl-PDZ3 or ICG-001 before cell death was assessed. Error bars show mean ± SEM from 11 patient samples; *$p < 0.05$, **$p < 0.01$, ***$p < 0.001$, ****$p < 0.0001$

from DAPT-treated mice compared to vehicle control mice (Fig. 7b and Supplementary Figure 12), similar to our in vitro observation (Fig. 2g). These data indicate that this treatment schedule targeted Notch2 on BMSCs in vivo. Assessment of C1q transcripts in sorted Sca1[+] BMSCs showed a reduction of mRNA levels by DAPT treatment (Fig. 7c), indicative of Notch inhibition. Notably, DAPT treatment reduced the number of tumour cells in the bone marrow by 50% (Fig. 7d). To assess whether DAPT treatment had an impact on cell proliferation, we labelled tumour cells with Far-Red CellTracker before transplantation. The progressive decay of the dye with each cell division allows tracking of cell proliferation. At 3 days after transplantation, mice were treated for 3 days with DAPT (Fig. 7e). Analysis of bone marrow-resident tumour cells showed that DAPT treatment delayed cell proliferation based on an increased staining for Far-Red CellTracker compared to vehicle control mice (Fig. 7f, g). Lastly, to translate these observations into human CLL, we reconstituted primary CLL cells from 4 individual, untreated patients in NSG mice. Notably, 3 of these patients did not carry Notch1 mutations (mutation status for 1 patient was not assessed). Peripheral blood mononuclear cells (PBMCs) from each patient were injected intravenously in 8 mice

per patient as previously described[29]. To allow time for CLL cells to home to lymphoid organs before therapy, mice were treated after 5 days once daily with DAPT, receiving a total of 10 doses over the course of 12 days (Fig. 7h). At 4 weeks after engraftment, mice were killed and lymphoma burden was assessed by flow cytometry and histology. Haematoxylin and eosin (H&E) staining showed multifocal infiltration of neoplastic lymphocytes in the splenic parenchyma, interspersed in areas of extramedullary haematopoiesis, the latter a normal feature of murine spleen. Mitotic figures were present within the lymphoid infiltrates. Staining of spleens with anti-human CD20 showed fewer CLL cells in DAPT-treated mice compared to control mice (Fig. 7i). For quantification, the degree of CLL infiltration was assessed by flow cytometry, confirming that DAPT treatment impaired splenic engraftment in the CLL-PDX model (Fig. 7j).

Next, we investigated the expression of activated, dephosphorylated β-catenin in human lymph node biopsies from CLL patients. We found nuclear expression of β-catenin throughout the infiltrated lymph node in 6 out of 6 samples investigated (Fig. 8a and Supplementary Figure 9). In addition to immunofluorescence, we utilised immunohistochemistry (IHC) for β-catenin staining on human lymph node and bone marrow sections. We observed a fine,

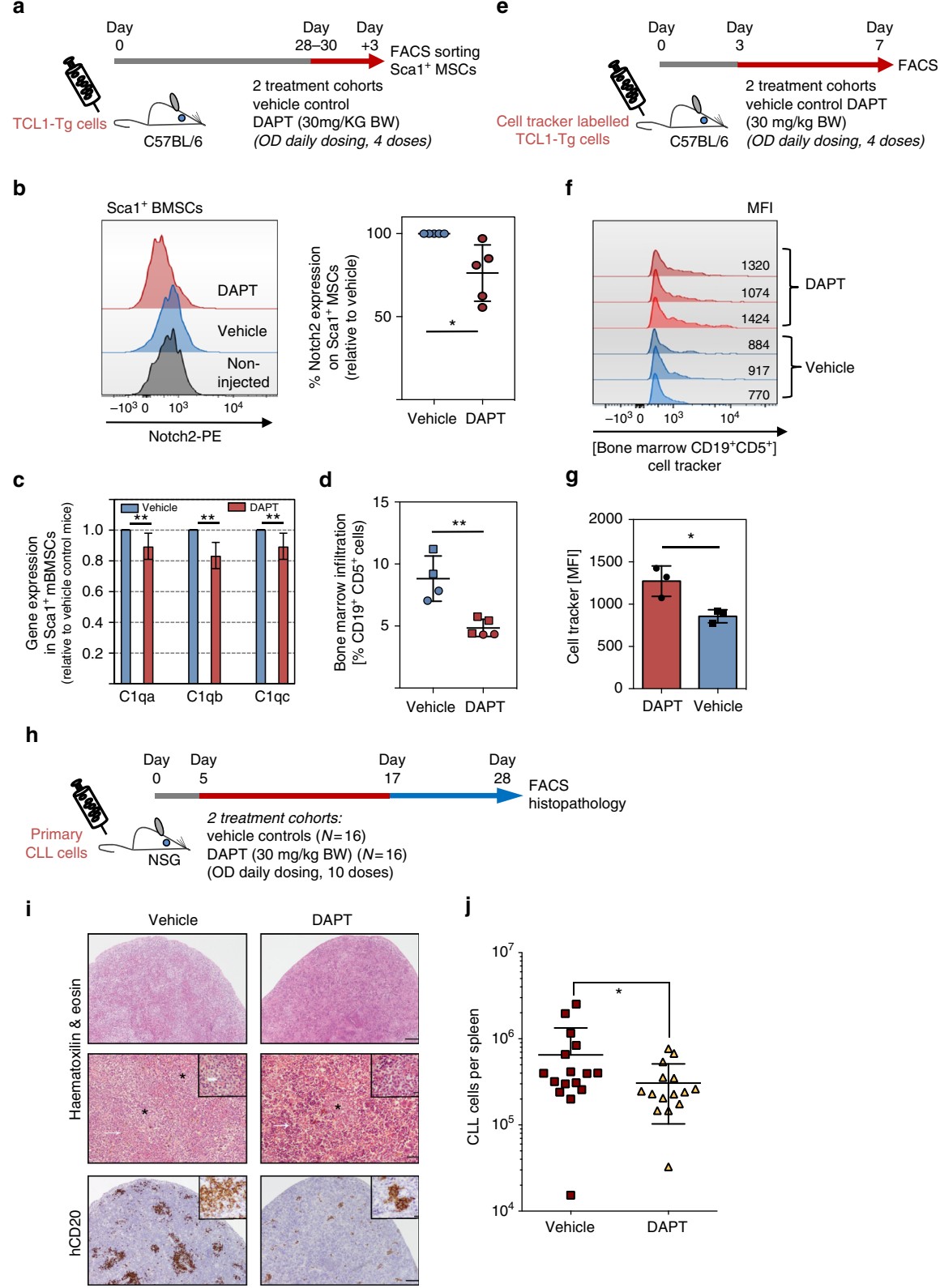

perinuclear dot positivity within the CLL cells across all tissue sites in all samples investigated. Notably, within tissue sections, the degree of positive staining varied from cells with an absent expression of β-catenin to cells, which showed a brighter staining (Supplementary Figure 10). These data suggested that Wnt signalling is active in CLL cells in tissues.

Lastly, we tested whether β-catenin and N-cadherin were similarly regulated by contact to BMSCs in B cells from other malignancies. Co-culture of malignant B cells from a leukaemic patient with diffuse large B-cell lymphoma (DLBCL) on EL08-1D2 cells induced a strong up-regulation of β-catenin and minimal N-cadherin expression. Two out of three high-grade

**Fig. 7** Notch targeting in vivo. **a** Schematic representation of the experimental design: mice transplanted with $40 \times 10^6$ splenocytes from diseased $TCL1^+$-$Tg$ mice were treated after 4 weeks with 4 consecutive doses of 30 mg/kg DAPT or vehicle control (i.p. injection; once daily). **b** Notch2 expression analysis in bone marrow stromal Sca1$^+$ cells of transplanted mice, treated with DAPT (ruby circles) or vehicle control (blue circles); ($n = 5$ for each treatment group; two different primary tumours were transplanted). Relative quantification of the MFI of treated versus untreated mice is shown on the right panel, *$p < 0.05$. **c** RT-qPCR analysis of the C1q complex in BMSC-Sca1$^+$ cells of mice treated with DAPT, normalised to the expression in mice treated with vehicle. Because of low RNA yield, Sca1$^+$ cells were sorted from three individual mice and then combined for RNA extraction. Shown is the mean ± standard deviation of three technical repeats. **d** Quantification of bone marrow CD19$^+$CD5$^+$ cells of mice treated with DAPT (red symbols) or vehicle control (blue symbols) using flow cytometry ($n = 5$ per treatment group, two different primary tumours were transplanted); **$p < 0.01$. **e** Schematic representation of the experimental design. **f** Far-Red CellTracker fluorescence of bone marrow-derived CD5$^+$CD19$^+$ cells, 7 days following transplantation ($n = 3$ per treatment group). **g** Quantification of the MFI of treated and untreated mice shown in (**f**); *$p < 0.05$. **h** Schematic scheme of the CLL-PDX model. Freshly isolated PBMCs from four untreated CLL patients were injected intravenously into four mice respectively. At 5 days after transplantation, mice were treated once daily with DAPT at a dose of 30 mg/kg body weight or vehicle control (i.p. injection). Mice received a total of 10 doses over the course of 12 days. **i** Lymphoma burden in the spleen was assessed by flow cytometry (staining for CD5$^+$, CD19$^+$ cells) and histology. The thin white arrow indicates areas of extramedullary haematopoiesis; thick white arrow: mitotic figure. *Cells with morphology and CD20 staining consistent with neoplastic cells. Scale bars: lines 1 and 3 = 300 μm; line 2 = 100 μm, insert in line 2 = 30 μm. **j** Number of splenic CD19$^+$CD5$^+$ presents in mice treated with DAPT (yellow triangles) or vehicle control (ruby squares); **$p < 0.01$

B-cell lymphoma (hgNHL) lines showed stroma-dependent stabilisation of β-catenin without significant changes in N-cadherin expression, suggesting that the regulation of N-cadherin is different in CLL and hgNHL cells. Primary malignant B cells from patients with Mantle cell lymphoma (MCL) exhibited a similar regulation of β-catenin and N-cadherin as CLL cells (Fig. 8b). These data indicate that BMSCs can contribute to canonical Wnt signalling in numerous B-cell malignancies.

## Discussion

The contribution of the tumour microenvironment to disease development, maintenance and progression has increasingly been acknowledged in recent years. Here we describe that the cross-activation of CLL cells and BMSCs results in Notch2 activation in stromal cells, which is in turn required for the activation of Wnt signalling in malignant B cells (Fig. 8c). The contribution of de-regulated Wnt/β-catenin pathway to tumourigenesis of solid tumours and haematological malignancies is well established through in vitro and in vivo studies[30]. In CLL, activation of the canonical Wnt pathway was reported by several groups, based on the constitutive over-expression of Wnt and dishevelled proteins and β-catenin target genes[24,31,32]. DNA hypermethylation of Wnt inhibitors is one mechanism reported to underlie this activation of Wnt signalling[33,34]. More recently, next-generation sequencing identified somatic mutations in multiple genes regulating β-catenin expression in altogether 14% of patients[5]. Our data indicate that contact to BMSCs in the microenvironment is suf-ficient to activate canonical Wnt signalling in malignant B cells. Since β-catenin stabilisation in CLL cells cultured on BMSCs is extremely robust and was observed in all our samples without exceptions, we conclude that the activation of the canonical Wnt pathway can occur in the absence of activating somatic muta-tions. An important question is how β-catenin contributes to disease initiation and progression. This issue was previously addressed, using small molecule inhibitors or gene silencing and data indicate that the survival of malignant B cells directly depends on Wnt signalling[5,6]. In agreement with this, our results demonstrate that small molecule inhibitors of Wnt signalling can overcome the protection of CLL cells by stromal cells in vitro. However, these studies only allow for drawing of limited con-clusions about the role of β-catenin in vivo. Since reported Wnt/β-catenin target genes supposedly include *Myc* and *cyclin D1*, it is reasonable to speculate that this pathway is also important for cell proliferation and disease progression in CLL. Indeed, in a transplantation assay, we also noticed reduced cell proliferation of Tcl-1-driven tumours in vivo. Further evidence in support of this idea originates from the $E\mu$-$Tcl1$ mouse model crossed into $Fzd6$

knockout mice, demonstrating that the Fzd6 receptor for Wnt ligands is up-regulated during B-cell leukaemogenesis[7], similar to primary human CLL cells[24]. Lack of Fzd6 significantly delayed disease progression, associated with a lack of β-catenin expression in malignant B cells.

Our data further advance the mechanistic understanding of canonical Wnt signalling in CLL by identifying Notch2, activated through CLL contact in mesenchymal stromal cells, as a key regulator of β-catenin in malignant B cells. This strict dependency of Notch2 was an unexpected and remarkable finding as BMSCs constitutively express all four Notch receptors. Notch1 and Notch2 are structurally similar with a 52% homology between the two proteins (56% for human proteins). However, Notch1 (or any other Notch receptor) was unable to compensate for the loss of Notch2 in CLL-activated BMSCs with regard to activating Wnt signalling in malignant B cells. In line with this observation, we found loss of cleaved Notch1 in co-cultured BMSCs, in striking contrast to Notch2. This raises the question of how CLL cells preferably activate Notch2 signalling. One possibility is that Notch ligands bind with different signal strength to individual Notch receptors, thereby defining signal specificity[35,36]. In addi-tion, post-translational modifications of the extracellular domain of Notch have been identified as another mechanism of regulating Notch–ligand interactions[19]. In addition, CLL cells may provide conditions to BMSCs permitting the recently suggested form of *cis*-activation of Notch signalling[37].

It is noteworthy that the degree of dependency on stromal Notch2 for β-catenin stabilisation in CLL cells varied between samples and we found that 25% of all primary CLL samples did not depend on Notch2 activity to express β-catenin in co-culture. Although this is not unexpected given the heterogeneity of the disease, to date we cannot explain this finding. We hypothesise that somatic mutations in Wnt signalling genes, present in a subset of patients[5], may overcome the dependency on stromal Notch2, while still being responsive to Notch2-independent, BMSC-mediated regulation of the Wnt pathway.

Gene expression profiles indicate that Notch2 signalling is constitutively active and regulates gene expression in BMSCs. In line with this observation, cleaved Notch2 was detectable in all primary (human and mouse) BMSCs in the absence of CLL cells. Contact to CLL cells induced substantial, Notch2-dependent changes in gene expression. The majority of these regulated genes were suppressed by Notch2, in keeping with the observation that Notch signalling can be largely repressive. Ultimately, these changes are required for the full stabilisation of β-catenin in malignant B cells. Our experiments identified several, Notch2-dependent soluble factors, which can contribute to the

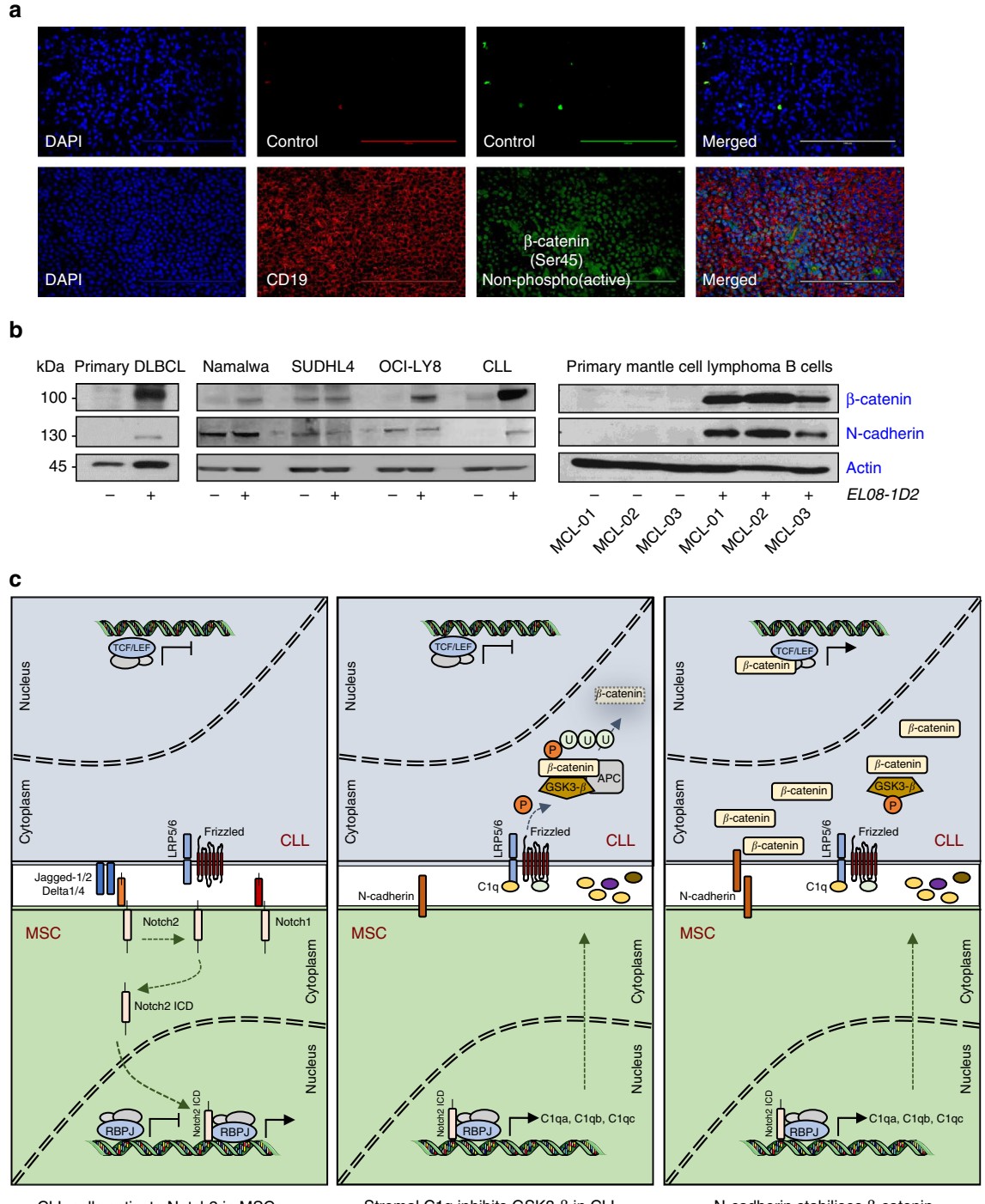

**Fig. 8 a** β-Catenin was assessed in lymph node sections of CLL patients by microscopy. A specific non-phospho-antibody for β-catenin was used to detect the active form of the protein. One representative result from six different patients is shown. Scale bar = 100 μm. **b** β-Catenin and N-cadherin expression in primary B cells from a patient with leukaemic DLBCL (left panel) or from three patients diagnosed with Mantle cell lymphoma (MCL) after 5 days of co-culture on EL08-1D2 cells (right panel). High-grade NHL cell lines (Namalwa = Burkitt lymphoma; SU-DHL-4 and OCI-LY8 = DLBCL) were cultured on stromal cells for 48 h (middle panel). **c** Schematic presentation of the mutual activation of BMSCs and CLL cells. (Left) CLL cells induce Notch2 activation in BMSCs. (Middle) Stromal Notch2 in turn regulates the expression of complement C1q and other soluble factors, required for the inhibition of GSK3-β and stabilisation of β-catenin in malignant B cells. (Right) In addition, up-regulated N-cadherin in CLL cells interacts with β-catenin and further contributes to its stabilisation. This figure was partly produced using the Smart Servier Medical Art, available from https://smart.servier.com/image-set-download/ and licensed under a Creative Common Attribution 3.0 Generic License. http://smart.servier.com/

stabilisation of β-catenin in CLL cells. Of these, complement C1q appears to play an important role for the inhibition of GSK3-β in malignant B cells. Beside its role in innate immunity, it was recently demonstrated that C1q, produced by stromal cells, has independent cancer-promoting activities in solid tumours[38]. In support of this idea, sera from CLL patients were found to contain higher C1q levels than normal controls[39]. Mechanistically, it was shown that this activation can occur upon binding of C1q to Fz receptors and subsequently C1s becomes activated. Cleavage of LRP5/6 by C1s then activates canonical Wnt signalling[26]. Our data demonstrate that the addition of C1q to CLL stroma co-cultures can compensate for Notch2 deficiency in BMSCs with regard to Wnt activation. Notably, C1q had no effect on β-catenin stabilisation in CLL mono-cultures, indicating that stromal cells provide additional, essential signals to activate this pathway (e.g., C1s, N-cadherin).

In addition to the C1q complex, the Wnt inhibitors *Sfrp4* and *Wif1* were up-regulated in Notch2-deficient stromal cells and their Notch2-mediated suppression in wild-type cells may further facilitate Wnt activation by BMSCs. Therefore, it is reasonable to assume that numerous factors act in a collaborative manner to stabilise β-catenin. The contribution of an individual factor to the activation of Wnt signalling will likely depend on its local concentration and accessibility in the niche.

Notably, conditioned media from CLL-MSC co-cultures lack the ability to vigorously stabilise β-catenin, indicating that Notch2 must co-regulate membrane-tethered proteins required for Wnt activation. We hypothesised that N-cadherin may be one of these factors, since β-catenin can bind tightly to type-I cadherins. Ablation of N-cadherin in BMSCs indeed significantly reduces N-cadherin and β-catenin expression in CLL cells. Plasma membrane profiling of BMSCs revealed several Notch2-regulated proteins expressed in stromal cells, which could be required for cell adhesion and also affect both the activation of stromal and CLL cells. Importantly, ablation of Notch2 from stromal cells significantly inhibited the induction of N-cadherin in CLL cells. Since N-cadherin and β-catenin expression, induced by BMSCs, follow a similar kinetic, it is reasonable to speculate that N-cadherin transcription is directly regulated by β-catenin, constituting a positive feed-back loop to further enhance β-catenin expression in tissues.

In addition to these mechanisms, Notch2-dependent alterations in the composition of extracellular matrix may affect integrin signalling in CLL cells and tissue stiffness. The latter effect was shown to induce aberrant mechanotransduction, leading to β-catenin stabilisation through disintegration of the deconstruction complex[40].

The identification of Notch2-regulated activation of Wnt in malignant B cells raises the question of whether this signalling pathway constitutes a meaningful target for therapeutic interventions. Our data demonstrate that treatment with the γ-secretase inhibitor DAPT significantly impairs engraftment in a CLL-PDX model. This clearly establishes γ-secretase as a therapeutic target in CLL. Notch proteins are prominent targets of γ-secretase and our in vitro experiments suggest that inhibition of Notch2 activity on stromal cells should be a contributing factor to the overall effect of DAPT. However, other effects such as direct inhibition of Notch signalling on CLL cells and interference with other γ-secretase-dependent pathways surely play additional roles. However, the net effect of γ-secretase inhibition in vivo resulted in reduced cell proliferation and clearly diminished engraftment.

In spite of the recent clinical success with B-cell receptor (BCR) inhibitors and B-cell lymphoma-2 (Bcl-2) antagonists, the disease still remains incurable due to residual cells surviving in protective niches. Therefore, treatment of minimal residual disease (MRD)

remains an on-going clinical objective with the aim to fully eradicate the disease. We believe that Notch2 microenvironment-mediated activation of β-catenin can be relevant for MRD in protective niches in the bone marrow and other lymphoid organs. Our analyses of β-catenin in sections from CLL patients indicate that a substantial fraction of cells harbours activated, nuclear β-catenin. These data contrast another report showing absence of nuclear β-catenin in bone marrow sections, which may be related to different antibodies used for these analyses. The degree of β-catenin activation by stroma-derived signals certainly depends on the duration of these cell–cell interactions in vivo as well as on the tissue concentrations of soluble, Wnt-regulating factors. It remains to be experimentally addressed to what extend BMSC-induced β-catenin regulates gene expression in tissues. For this, we currently perform chromatin immunoprecipitation-sequencing experiments on stroma-activated CLL cells.

In support of the idea that Wnt activation is characteristic for the niche in CLL, it was demonstrated that BMSC-mediated activation of the Wnt pathway protects CML-progenitors from TKI-treatment[27]. Similarly, microenvironment-mediated stabilisation of β-catenin increases drug resistance in acute leukaemias[41]. Therefore, Notch- or Wnt inhibitors may be best used in combination therapies in B-cell malignancies. Alternatively, single agent therapies may have the potential to achieve deeper, MDR-negative, remission when given as consolidation treatment. Since we also identified BMSC-mediated stabilisation of β-catenin in MCL and hgNHL B cells, such treatment modalities may be relevant also in other B-cell malignancies.

## Methods

**Primary cells and cell culture**. After informed patients' consent and in accordance with the Helsinki declaration, peripheral blood was obtained from patients with a diagnosis of CLL, DLBCL and MCL. These studies were approved by the local ethical committee of the Technical University Munich (project number 1894/07) and by the Cambridgeshire Research Ethics Committee (07/MRE05/44). Human BMSCs were collected from individuals who underwent bone marrow aspiration for diagnostic purposes and in whom subsequently a haematological disease was ruled out. Patients consented to the use of this material for research purposes. Murine BMSCs were established from femora and tibiae of 4- to 8-week-old mice accordingly. PBMCs were isolated from heparinised blood samples from patients by centrifugation over a Ficoll-Hypaque layer (PAN-Biotech, Aidenbach, Germany). If the content of $CD5^+CD19^+$ cells was less than 90%, CLL cells were further purified by incubating cells with anti-CD2 and anti-CD14 magnetic beads (Dynabeads, Thermo Scientific, Winsford, UK) according to the manufacturer's instructions. After purification, B cells from CLL patients were generally >95% pure as assessed by flow cytometry. Samples for RNA-seq were >98% pure. After harvest, malignant B cells were either frozen down as viable cells or cultured in RPMI 1640 (Gibco, Darmstadt, Germany) supplemented with 10% foetal calf serum, penicillin/streptomycin 50 U/ml, Na-pyruvate 1 mM, L-glutamine 2 mM, L-asparagine 20 mg/ml, 2-mercaptoethanol 0.05 mM, HEPES 10 mM and minimum essential medium (MEM) non-essential amino acids (Gibco). No differences were observed between thawed cells and fresh cells with regards to the results presented in this study. EL08-1D2 and all primary mouse bone marrow stromal cell cultures were cultured in MEM Alpha+GlutaMAX medium (ThermoFisher Scientific, Winsford, UK) supplemented with 10% foetal calf serum (Gibco), 10% horse serum (Sigma-Aldrich, Dorset, UK), 10 μM 2- mercaptoethanol and 1% penicillin/streptomycin (Gibco). For co-culture experiments, $15 \times 10^4$ EL08-1D2 cell were seeded in a 6 multi-well plate 24 h before the addition of primary CLL cells at a concentration of $10^6$ cells/ml (equalling $3 \times 10^6$ total primary CLL cells).

Cell lines used for the experiments depicted in Fig. 8b were obtained from Dr Daniel Hodson as a gift from the Staudt lab (National Cancer Institute, Bethesda, USA). Cell lines and primary stromal cells were frequently tested for mycoplasma to ensure contamination-free conditions.

**Transwell co-culture assays**. EL08-1D2 or primary human bone marrow stromal cell were seeded in 24 multi-well plates. After 24 h, purified primary CLL cells were added in 1 μm transwell insert (Sarstedt, Nümbrecht, Germany) at the concentration of $10^6$/ml.

**Reagents**. Nuclear-cytoplasmic protein separation was performed with NE-PER™ Nuclear and Cytoplasmic Extraction kit (ThermoFisher Scientific, Winsford, UK) according to the manufacturer's instructions.

The agents DAPT, ICG-001 and XAV939 were purchased from Selleckchem (Newmarket, UK). Dvl-PDZ 3 was purchased from Merck Millipore (Billerica, MA, USA).

**Antibodies**. A list of all antibodies used in the study, including catalogue numbers and dilution, can be found in Supplementary Table 4. Uncropped images of the most important western blots are provided in Supplementary Figure 13.

**Expression analysis/qPCR**. Total RNA was isolated using the RNeasy Mini Kit (Qiagen, Manchester, UK), and complementary DNA (cDNA) was obtained using the qScript$^{TM}$ cDNA SuperMix kit (QuantaBio, Beverly, MA, USA). Quantitative reverse-transcription polymerase chain reaction (RT-qPCR) was performed on isolated mRNA using the fast SYBR reagents and a ViiA 7 real-time PCR system (Life Technologies, Warrington, UK). Gene target expression levels were normalised using *GAPDH*, and values are represented as fold change relative to control using the ΔΔCt method. Primer sequences for quantitative RT-PCR can be found in supplementary table 5.

**Immunofluorescence**. EL08-1D2 and CLL cells, after co-culture on glass coverslips coated with poly-L-lysine for 24 h, were fixed and permeabilised using ice-cold methanol. After blocking the sections with 4% normal goat serum in phosphate-buffered saline (PBS) at room temperature for 30 min, β-catenin antibodies, β-Catenin (D10A8) XP® Rabbit, Non-phospho (Active) β-Catenin (Ser45) (D2U8Y) XP® Rabbit and Non-phospho (Active) β-Catenin (Ser33/37/Thr41) (D13A1) Rabbit (Cell Signaling Technology, Leiden, The Netherlands) were incubated at 4 °C overnight followed by Anti-rabbit IgG (H+L), F(ab')2 Fragment (Alexa Fluor® 488 Conjugate) (Cell Signaling Technology, Leiden, The Netherlands).

Lymph node tissue samples were rehydrated and the Antigen Retrieval Reagent-Acidic buffer (R&D Systems) was used according to the manufacture's instruction. After blocking the sections with 4% normal horse serum in PBS at room temperature for 30 min, Non-phospho (Active) β-Catenin (Ser45) (D2U8Y) XP® Rabbit (Cell Signaling Technology, Leiden, The Netherlands) and CD19 [2E2B6B10] (Abcam, Cambridge, UK) were incubated at 4 °C overnight followed by Anti-rabbit IgG (H+L), F(ab')2 Fragment (Alexa Fluor® 488 Conjugate) (Cell Signaling Technology, Leiden, The Netherlands) and Goat Anti-Mouse IgG H&L (Cy5 ®) (Abcam, Cambridge, UK).

Nuclei were stained with ProLong™ Gold Antifade Mountant with 4',6-diamidino-2-phenylindole (DAPI; ThermoFisher Scientific, Winsford, UK) and images were captured using the Leica TCS Sp5 confocal (Leica Microsystems, IL, USA) or the EVOS FL Cell Imaging System (ThermoFisher Scientific, Winsford, UK). The images were processed using ImageJ software.

**Multiplex protein signalling array**. Primary CLL cells were mono-cultured or co-cultured on EL08-1D2 cells for 6 h. Phosphorylation of common signalling pathways was detected using The PathScan® Intracellular Signaling Array Kit (Chemiluminescent Readout) (Cell Signaling Technology, Leiden, The Netherlands) according to the manufacturer's instruction. The image of the slide was captured with the Azure Biosystem c300 (Dublin, CA, USA) digital imaging system.

**Flow cytometry**. Multicolour flow cytometric analysis of Nestin$^+$, endothelial and osteoblast compartments was performed at 2 months of age as previously reported[42]. In brief, Nestin$^+$ mesenchymal cells and other stromal cell populations were isolated from femur and tibia. Bones were crushed in a mortar with a pestle and digested with 2 ml of collagenase (Stem Cell Technologies, 07092) and 1 mg/ml Collagenase IV (Sigma) at 37 °C in strong agitation for 30 min. Cells were then filtered through a 40 μm mesh and red blood cells lysed with 1× Pharmlyse (BD) for 10 min on ice.

Apoptosis was detected using the Annexin-V Apoptosis Detection Kit I (eBioscience, Winsford, UK). Samples were acquired on a LSRII flow cytometer (BD Biosciences, Oxford, UK) and analysed using FlowJo software (Tree Star).

Antibodies for fluorescence-activated cell sorting analysis are listed in Supplementary Table 6.

**CRISPR/Cas9 plasmids**. Single-guide RNA (sgRNA) sequences were cloned into lentiCRISPRv2 (the individual sequences are listed in supplementary table 7). Lentiviral infections of EL08-1D2 and murine bone marrow stromal cells with the specific sgRNA constructs were performed. Following 48 h of puromycin selection (2 μg/ml), cells were negatively sorted for Notch2 expression and cultured for further experiments.

**Mass spectrometry**. Plasma membrane profiling was performed as described previously[28]. Peptides were subsequently labelled with TMT reagents (Thermo Fisher Scientific, Winsford, UK), pooled and cleaned up using a SEP-PAK C18 cartridge (Waters) prior to high pH RP fractionation as previously described[43]. High pH fractions were pooled orthogonally into 12 samples for analysis by liquid chromatography–mass spectrometry on an Orbitrap Fusion (Thermo Fisher Scientific, Winsford, UK) utilising synchronous precursor selection mode to isolate reporter ions essentially as previously described[43]. Data were searched using the

MASCOT (Matrix Science, UK) search node within Proteome Discoverer v2.1 (Thermo Fisher Scientific, Winsford, UK). The database used was the SwissProt Mouse Reference Proteome including an appended database of common contaminants. Statistical differences between replicate groups were assessed using an implementation of LIMMA within the R environment including Benjamini–Hochberg correction for multiple hypothesis testing. The resulting $p/q$ values are reported.

**Microarray and RNA-seq**. Microarray analyses were performed using the Affymetrix GeneChip platform employing the Express Kit protocol for sample preparation and microarray hybridisation (Affymetrix, Santa Clara, CA). Total RNA (200 ng) was converted into biotinylated cRNA, purified, fragmented and hybridised to MG-430_2.0 microarrays (Affymetrix). The arrays were washed and stained according to the manufacturer's recommendation and finally scanned in a GeneChip scanner 3000 (Affymetrix) with G7 update. Array images were processed using the RMA algorithm implemented in PartekGS (Partek, St. Louis, MO).

RNA samples for RNA-seq were measured by Nanodrop spectrophotometry and then analysed on the Agilent Bioanalyzer RNA Nano chip. Samples (100 ng total RNA) were then processed for next-generation sequencing using the Ovation Human FFPE RNA-Seq Multiplex System Kit (NuGEN, 0340-32, 0341–32). During library preparation, a unique method (InDA-C) was used to deplete unwanted reads derived from ribosomal RNA. The size distribution of the resulting libraries was analysed on Agilent Bioanalyzer HS DNA chips. All libraries showed the expected size profile and sufficient yield. Since the kit contains a limited number of barcodes, two library pools containing 9 samples each were generated for sequencing and quantified using the NEB Library Quant kit, a SYBRgreen-based qPCR method. The sequencing of the library pool was performed on a HiSeq 2500 High Output flow cell in the paired-end mode. For quantification of RNAs, qm-filtered alignments were imported into PartekGS. Subsequently, differentially expressed genes were identified using the analysis of variance (ANOVA) test implemented in PartekGS. The stepup method for multiple testing correction was applied to generate corrected $p$ values.

**CLL-PDX model**. These animal studies were performed in accordance with state law and were approved by the local ethical committee of the University Essen-Duisburg, Germany (project numbers 14-6080-BO and G1124/10). NOD.Cg-PrkdcscidIl2rgtm1Wjl/SzJ (NSG) (5- to 10-week-old) mice were used as recipients for transplantation of primary human CLL cells as described previously[29]. A total of $10^8$ mononuclear cells were intravenously transplanted into NSG mice 24 h after treatment with Busulfan (30 mg/kg) (Selleckchem, Newmarket, UK). Mice were killed 4 weeks after transplantation. For analysis of engraftment, one half of the murine spleen was fixed in buffered 3.7% formalin and afterwards embedded in paraffin for further immunohistochemical analyses. The other half of the spleen was mechanically homogenised, pressed through 70 μm nylon sieves (BD Biosciences, Oxford, UK), and suspended in PBS for further analysis by flow cytometry.

**Eμ-tcl1tg mice**. These animal studies have been regulated under the Animals (Scientific Procedures) Act 1986 Amendment Regulations 2012 following ethical review by the University of Cambridge Animal Welfare and Ethical Review Body (AWERB-PPL number P846C00DB). *Eμ-Tcl1tg* mice[44] were kindly provided by Dr. Carlo Croce. Tumour cells from moribund *Eμ-Tcl1tg* were labelled with CellTrace™ Far-Red Cell Proliferation Kit (ThermoFisher Scientific, Winsford, UK) according to the manufacturer's instructions. $40 \times 10^6$ cells were then transplanted intraperitoneally into 6- to 8-week-old syngeneic C57BL/6 mice. Mice were killed 7 days after transplantation.

**Mouse immunohistochemistry**. H&E-stained histological sections from 33 individual mice were examined by a board-certified veterinary pathologist blinded to the identification codes indicating whether the mouse had received vehicle or DAPT. Immunohistochemical staining for human CD20 (1:400; RB-9013; rabbit polyclonal antibody; Thermo Fisher Scientific, Winsford, UK) was carried out following de-paraffinization and antigen retrieval (20 min at 90 °C) in a PT Link, Pre-Treatment Module for Tissue Specimens utilising Dako Envision Flex Target Antigen Retrieval Solution High pH (both from Dako Agilent Technologies, Stockport, UK). A routine immunohistochemistry protocol employing an automated system (Dako Autostainer) was followed. Negative control slides were treated with isotype- and species-matched immunoglobulins. Negative control tissue from a mouse without splenic lesions was obtained from the archives of the Department of Veterinary Medicine.

**Human immunohistochemistry**. Immunohistochemistry was performed on whole formalin-fixed, paraffin-embedded (FFPE) sections on both lymph node and bone marrow tissues. Reactions were performed on Leica Bond-III (Leica Microsystems, IL, USA). CD20 (monoclonal mouse anti-human CD20, clone L26, Dako) antibody was diluted 1:100 with 20 min heat retrieval in 9.0 pH buffer, while the β-catenin (ready to use primary β-catenin antibody, clone 17C2, Leica) antibody was pre-diluted with 20 min heat retrieval in 6.9 pH buffer.

**Statistical analyses**. Statistical analyses of the results were performed by one-way ANOVA followed by Student's *t*-test. Throughout the study, statistical significance was defined as ****$p < 0.0001$, ***$p < 0.001$, **$p < 0.01$, *$p < 0.05$ or ns (statistically non significant).

## Data availability

The authors declare that the data supporting the findings of this study are available within the paper and its supplementary information. The gene expression profile data have been deposited in the GEO database under accession numbers GSE99724 (Microarrays, Fig. 1) and GSE99614 (RNA-seq, Fig. 3). The mass spectrometry proteomics data have been deposited to the ProteomeXchange Consortium via the PRIDE partner repository with the dataset identifier PXD010401 and https://doi.org/10.6019/PXD010401. All other remaining data are available within the article and Supplementary Files, or available from the authors upon request.

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

## Acknowledgements

We would like to express our deepest gratitude to patients who donated blood for research purposes. In particular, we thank Dr. Joanna Baxter and her team for enrolling patients in these studies. We are also grateful for the help of Dr. Marco Galardini for his bioinformatics support. This work was funded by Cancer Research UK (CRUK; C49940/A17480) and by the Deutsche Forschungsgemeinschaft (DFG, FOR 2033 (projects B3, B6 and B7)). I.R. is a senior CRUK fellow.

## Author contributions

M.Ma., F.G., A.M., T.A., M.O., A.S. and G.L-G. planned, performed and analysed all in vitro experiments. Nestin^GFP mice were provided by S.M-F; L.K-H. performed and analysed microarray and RNA-seq experiments. P.J.L. and J.C.W. ran and analysed plasma membrane profiling mass spectrometry experiments. J.D., M.Mö. and K.H. performed and analysed the PDX experiments. L.R-B. provided IHC tissue staining from lymph node and bone marrow biopsies. EL08-1D2 cells were provided by R.A.J.O. Notch2^f/f mice and BMSCs from these mice were obtained from U.S. B-cell lymphoma

lines were originally obtained from L.Staudt and provided by D.H. M.S-S., C.P., D.H. and I.R. contributed conceptually to this work. M.S.S. and I.R. wrote the manuscript.

## Additional information

**Competing interests:** The authors declare no competing interests.

