## [Peer Review File · Nature Communications]

Reviewers' Comments:

Reviewer #1:

Remarks to the Author:

The work by Mangolini et al. aims at underlying the critical role for Notch and Wnt signaling in microenvironment-dependent survival of chronic lymphocytic leukemia (CLL) cells. The Authors used bioinformatics, genetics, and in vitro and in vivo tools to demonstrate that Notch2 and β -catenin pathways in CLL are the main actors of the pro-survival bidirectional crosstalk between stromal cells and leukemic cells. The main outcome of this study is that small molecule inhibitors of Notch and Wnt/ β -catenin may have an anti-tumoral role in CLL. The study is well designed and the manuscript is well written; however, some issues need to be further clarified.

1-Novelty

Although the role for Notch pathway in promoting stromal cell-mediated pro-survival signaling has been described elsewhere, the study gives new interesting insights about the mechanisms, describing a crosstalk between Notch and Wnt signalling. The most interesting findings are the in vivo activity of DAPT, confirming the anti-tumoral role of Notch inhibitors.

2-Expression of Notch receptors in MSCs

The Authors excluded a role for Notch receptors other than Notch2, demonstrating a reduction of Notch1 activation upon co-culture and poor expression of Notch3 and Notch4; the latter finding is inconsistent with previous papers clearly showing, through flow cytometry and western blot, that Notch 3 and Notch 4 are highly expressed by MSCs. The Authors should implement Western blot analysis by using anti-Notch3 and anti-Notch-4 antibodies capable of recognizing all the forms of each receptors, including cleaved (active) and full receptors.

Figure 2: the Authors show Notch expression in CLL samples and stromal cells, but Western Blot and flow cytometry are presented alternatively. The two techniques are complementary and should be used together for each set of experiments. In addition, to validate the expression of ICD1 and ICD2, the Authors should show the results of the analysis following cell exposure to a specific Notch inhibitor, such as DAPT.

3-Initial observations with EL08-1D2

- The Authors reported the initial observations using co-culture of 6 CLL samples on EL08-1D2. Regardless the demonstration of the supportive effects of EL08-1D2, human instead of murine MSCs should be preferably used to assess the interactions between leukemic and stromal cells, at least for this introductory experiment. Accordingly, in Figure 1, cocultures should be carried out with human MSCs to confirm the findings obtained with EL08-1D2

- Classic culture media, such as complete RPMI, are not adequate to support efficiently leukemic cell growth. The difference in expression between co-cultured cells and cells cultured alone could simply reflect a culture artefact, which could be prevented by using alternative culture media.

4-Therapeutic targets of the Notch2/ β -catenin crosstalk

The Authors stated that C1q addition rescued Notch2 deficiency; this concept should be explained better.

DAPT did not seem to affect the viability of CLL cells if cultured alone. Nwabo et al. (BCJ 2012) described CLL cell apoptosis by adding gamma-secretase inhibitors when cells were grown both alone and in co-culture. Similarly, Gandhirajan et al. (Neoplasia 2010) demonstrated that CLL cells cultured alone underwent apoptosis when treated with small molecule inhibitors of the Wnt/ β -catenin pathway. The Authors should try to explain these different findings.

MSCs physiologically express high levels of β -catenin that is necessary for their anti-apoptotic functions. Thus, one cannot exclude that Wnt/ β -catenin pathway inhibitors act indirectly by modulating the supportive properties of MSCs.

Reviewer #2:

Remarks to the Author:

In Mangolini et al., the authors describe several novel mechanisms that contribute to the observation that CLL B cells are more resistant to apoptosis when co-cultured with stromal cells derived from bone marrow. These mechanisms, elegantly deciphered through a combination of transcriptomic, proteomic, functional genetic, and chemical genetic approaches, substantially refine our understanding of tumor-microenvironment interactions and their contributions to cancer cell hallmarks. In their study, the authors discover that co-culture of bone marrow-derived stromal cells (BMSCs) with human CLL B cells leads to:

- 1) Upregulation of cell signaling and adhesion pathways in BMSCs
- 2) Increased phosphorylation of the WNT pathway member GSK3-beta in CLL cells
- 3) Decreased apoptosis activation in CLL cells that requires direct contact between BMSCs and CLL cells
- 4) Activation of Notch2 in BMSCs, leading to transcriptional upregulation of soluble cytokine factors such as C1q
- 5) C1q-mediated activation of the beta-catenin pathway in CLL cells through phosphorylation of GSK3-beta and stabilization of beta-catenin
- 6) Transcriptional upregulation of N-cadherin in CLL cells, leading to homotypic interactions with N-cadherin expressing BMSCs, leading to further stabilization of beta-catenin
- 7) Selective therapeutic effects of WNT pathway and Notch pathway inhibitors on CLL cells (when in BMSC co-culture)

The experiments the authors use to establish these conclusions are well-designed and well-controlled, and are relatively well-described. Publication of these results will provide useful information for the field, and do advance the current state of knowledge regarding tumor-stroma crosstalk pathways.

In order to strengthen the study, I feel that several minor improvements could be made, including:

- 1) A more complete description either in the methods or the results section of the transwell assay shown in Figure 1e. No mention of the technical details of this experiment is included in the methods. For example, after transwell cultures, what population of cells were analyzed for apoptosis? CLL cells only, or were they mixed with BSMCs? When CLL cells are co-cultured with BSMCs followed by apoptosis analysis, how is CLL-specific apoptosis identified? Is there staining for CD19 and Annexin V? (Supp Fig 5a does not seem to suggest this). Or are the cells purified by CD19 selection prior to flow analysis?
- 2) In Figure 3f, it should be noted in the legend why several genes are highlighted in red (soluble factors?).
- 3) In Figures 1b and 3d, there are two different y-axis – yet these are not described in the legend. Which axis refers to the histogram, and which axis refers to the line graph?
- 4) The authors show that N-cadherin expression in CLL cells is increased transcriptionally when in co-culture with BSMCs, and that this upregulation is dependent on N-cadherin+ BSMCs and is coincident with beta-catenin stabilization. Is the N-cadherin gene a direct target of TCF/LEF/B-cat complex? Or is this effect secondary to WNT pathway activation? Furthermore, does pharmacologic WNT pathway inhibition affect N-cadherin expression?
- 5) It is well-understood that Notch1 pathway activation is observed in a subset of CLL cases, either by mutation of the NOTCH1 gene or through other mechanisms. Thus, the activity of Notch pathway antagonists such as gamma-secretase inhibitors (eg DAPT) on CLL cell survival is likely a combination of effects on BSMCs and CLL cells themselves. The authors are sure to mention this

possibility in their Discussion, yet it would be informative to know what the contribution of BSMCs are to Notch signaling in CLL cells. Co-culture does not induce Notch1 cleavage/activation in BSMCs, but what about in CLL cells? While an in-depth study of the relative effects of DAPT on CLL versus BSMCs in vitro or in vivo is outside the scope of this study, it will nevertheless be very important to begin to understand this if it is to be pursued as a therapeutic strategy as the authors suggest it should. So, for example, in Figure 6d – of the 11 patient CLL samples used for the DAPT treatment study, how many are Notch1 ICD positive? Is there a difference in DAPT effects between Notch1 ICD+ and ICD- samples? If yes, this suggests dual effects on BSMCs and CLL cells are contributing to apoptosis. If no, this may suggest that gamma-secretase inhibition could be acting primarily through the novel mechanism described in this manuscript.

Reviewer #3:

Remarks to the Author:

Mangolini and coworkers use a variety of complementary approaches to arrive at and test a complicated model in which CLL cells expressing Notch ligands stimulate stromal cells expressing Notch2 to express C1q, which in turn activates Wnt signaling in CLL cells, leading to upregulation of N-cadherin and beta-catenin stabilization. The use of Notch2 knockout stromal cells clearly implicates Notch2 in the observed phenomena, and the stabilization of beta-catenin protein demonstrated in the culture system used is impressive. However, other aspects of the studies raise questions about the system and some of the reagents that are used as well as the cause and effect relationships proposed by the authors, muting enthusiasm for the manuscript as submitted.

Specific Comments

1. The authors' focus on CLL-to-stroma Notch signaling runs counter to the prevailing model in the field of stroma-to-CLL Notch signaling as a critical oncogenic mechanism in CLL, which is supported by significant prior literature, and by the strong genetic signature of selection for activating Notch receptor gene mutations in CLL and related small B cell lymphomas. Stroma-to-lymphocyte Notch signaling is also known to play an important role in sustaining non-neoplastic lymphocyte populations, such as thymic T cell progenitors, marginal zone B cells, and T-follicular helper cells. There are well described barriers that inhibit a given cell from simultaneously acting as a Notch ligand-presenting "sending" cell and Notch receptor-signal "receiving" cell, most notably the phenomenon of Notch receptor-ligand cis-inhibition. One can imagine ways in which this barrier might be overcome, but the authors do not discuss or address this issue at all.
2. Fig. 1. Important details of the co-culture assays are not stated in the methods, including whether experiments were done with freshly isolated or cryopreserved cells, and the total number (not just concentration) of cells added to a given well size.
3. Fig. 1. The results of the RNA-seq experiments presented in Fig. 1 are difficult to interpret, since the authors compare cells in low-density monoculture for 4 hr (unclear if this is after fresh isolation or after thawing) to cells that have been cultured for 48 hr in the presence of stromal cells. Differences in transcript levels could be due to any of a number of variables that are different between these two conditions (stromal contact, CLL-CLL contact, time of exposure to FCS and other media components etc). These multiple variables undoubtedly all contribute to the large number of genes whose expression is altered in cells cocultured with EL08-1D2 cells relative to control cells. What about RNA-seq on uncultured primary cells? In other words, to what extent are stromal cells changing gene expression versus maintaining gene expression that exists prior to culture? The same question pertains to the proteomic experiments described in figure 1. The finding in figure 1E that direct contact with stromal cells supports survival of CLL cells in low-density culture has been shown repeatedly in the prior literature and is not novel. These concerns make inclusion of these data, at least within anything other than supplementary data, questionable.

4. Figure 2A. Santa Cruz BT antibodies are notoriously unreliable (unlike Cell Signaling antibodies, which are trustworthy). For example, the JAG2 antibody used (H-143), if one looks at the SCBT website, appears to be cross-reacting with a band of ~200 kb in GM-CSF stimulated Jurkat cells, whereas the predicted size of full-length JAG2 is 133 kD (in fact, the SCBT website shows another JAG2 “specific” antibody that is detecting a protein of a completely different size in extracts prepared from a different cell type). In the western blot that is shown, no molecular masses are given, and there appear to be multiple cross-reactive polypeptides. The onus is on the authors to demonstrate that the bands shown on Western blot correspond to the proteins of interest.

5. Figure 2B-C. These figures show that NOTCH2 receptor expression on the authors’ stromal cell line of choice, EL08-1D2, is much higher (~10-fold) than in bone marrow stromal populations in vivo (this is made less noticeable by the authors’ use of a biexponential scale for Notch receptor flow in EL08-1D2, but not in the flow done on in vivo populations). This suggests that authors’ co-culture model may facilitate stromal Notch2-dependent effects that may not be relevant in vivo.

6. Figure 2D-G. A) The decrease in “NICD1” levels that occur in EL08-1D2 cells is misleading, since the antibody used in this blot will detect the Notch1 NTM and is not NICD1 specific. B) The identification of bands on the Western blots labeled NICD2 is presumptuous, since the antibody employed in these blots recognizes all Notch2 polypeptides containing the intracellular domain, including the furin processed NTM subunit. Additional proof is needed that NICD2 levels rise in co-cultured EL08-1D2, such as IF for Notch2 or subcellular fractionation.

7. Suppl Table 1 / Suppl Fig. 2B: If CLL cells are in fact activating Notch signaling in the stromal cells via stromal NOTCH2, we would expect to see increased expression of canonical genes that have been identified as direct Notch targets across many tissues, such as NRARP, increased in the Notch2 proficient versus Notch2 deficient (CLL activated) comparison, but these do not appear in the list of 2-fold upregulated genes. Hes1 does appear in Fig. S2B, but no scale is provided and it’s unclear if there is a large or significant increase. Does a western blot for Hes1 in CLL-activated NOTCH2 proficient vs NOTCH2 deficient stromal cells show a significant increase in the former (i.e. can Fig. 6C be replicated with stroma +/- NOTCH2 knockout rather than GSI)?

8. What, if anything, is known about the genotype of the CLL cells that were used to stimulate stromal cells? We are told in some experiments that they are not Notch mutated, but if anything else is known about the molecular-cytogenetic character of these samples, it should be provided for completeness.

9. Fig. 3. Notch2 has an important role in regulating the differentiation of certain bone marrow stromal cells, and consistent with this there are differences in gene expression between the Notch2 floxed and Notch2^{-/-} BMSCs. What are the genes/pathways that define these differences? Conversely, CLL cells clearly have large effects on Notch2^{-/-} BMSCs, based on PCA. What are the genes/pathways that define these differences? Arguably, these uncontrolled variables make it very difficult to infer precisely what the role of Notch2 is in this system. Other approaches, such as use of antibodies that specifically block Notch2 signaling, might yield more easily interpretable results and at a minimum would provide complementary information.

10. A potentially intriguing observation is the upregulation of beta-catenin in CLL cells by co-culture with Notch2-replete BMSCs and not by Notch2^{-/-} BMSCs. The experiments in Fig. 4A-B are not adequately described in the legends. If the comparison is between CLL cells kept in low-density culture for 5 days +/- stromal cells, the CLL in the no-stroma group may be largely apoptotic (see Fig. 1E); presumably this is not the case, but it needs to be spelled out. The 24 hr conditioned medium lane in Fig. 4C shows some evidence of beta-catenin upregulation by conditioned media (as the authors discuss later); the size of the effect should be clarified by providing a better Western blot, and the effect should be replicated across multiple CLL samples. Again, complementary experiments with acute perturbagens (GSI, blocking N2 antibody) would help to

build confidence in a direct link between Notch2 and beta-catenin stabilization. The experiments with CRISPR/Cas9 Notch2 knockouts are helpful to mitigate this concern, but the methods do not explain how these cells were isolated (single cell clones, pooled clones, etc.). Additional information is needed.

11. Fig. 4. The authors do not show whether the CLL beta-catenin induction effect is also mediated by mBMSC's +/- Notch2 deletion (only +/- EL08-1D2 co-culture data is shown). It is important to show that the stromal Notch2 / CLL beta catenin effect is not an idiosyncratic property of the EL08-1D2 cell line, which seems to have unusually high expression of NOTCH2.

12. Fig. 5A-E and I-J. These experiments establish that co-culture facilitates CLL N-cadherin expression, and that this effect and CLL beta-catenin expression are dependent on stromal N-cadherin and (in the co-culture system) on stromal Notch2. The authors should show that this effect is not simply due to increased CLL-CLL contacts facilitated by aggregation on stromal cells, by comparing stromal co-culture to high-density CLL monoculture, or culture of CLL cells in round-bottom plates that force aggregation of CLL cells.

13. Fig. 5F-H. These are interesting results, but they are confined to the EL08-1D2 cell line, which as discussed above, may not be representative of stromal cells that CLL cells are likely to encounter in vivo.

Figure 6.

14. Fig 6A. The prosurvival effect in CLL cells that is dependent on stromal NOTCH2 appears to be quite modest. Furthermore, the authors seem to have combined data from two different experiments (co-culture with cre-deleted Notch2 and Cas9-deleted Notch2 in MSC's) in order to make the figure, raising the question of whether either of the two original experiments on their own showed a significant effect. Could the difference in CLL survival on Notch2 +/- stroma be more impressive for that subset of CLL samples with a strong stromal NOTCH2 dependency for beta-catenin expression (per figure suppl 3e), or is there no correlation? Also, why was this experiment not also conducted with EL08-1D2 stromal cells +/- Notch2 deletion, the primary model used by the authors in Fig. 4 experiments?

15. Fig. 6D-E. The y-axis should be plotted as the absolute % apoptotic cells rather than apoptosis "relative to untreated cells" since this hides the baseline apoptosis values for the monoculture (which are likely very high). It may be best to simply remove the monoculture data from this figure. Statistical comparisons should be vs. untreated cells in the same culture system – it's unclear what is the biological significance of comparing relative difference in % apoptosis across culture systems with very different baseline levels of apoptosis.

16. The DAPT experiments in Fig. 7 are difficult to interpret because this treatment will block Notch signaling in the CLL cells as well as in the EL08-1D2 cells. Clearly, Notch signaling is important in these cells since there is selected pressure for Notch gain of function mutations in CLL. Similarly, the same confounding issue hangs over the in vivo PDX modeling work in which mice are treated with DAPT. To ignore this facet of the system seems an important oversight. Similarly, the authors never discuss possible Wnt signaling/beta-catenin expression in BMSCs; can effects on stromal cells be excluded in experiments with Wnt inhibitors?

17. In the description of the results in Fig. 7, the authors state that "the degree of infiltration was assessed by staining cells for human-CD45+, CD19+, CD5+ and flow cytometry, confirming that DAPT treatment impaired splenic engraftment in the CLL-PDX model (Figure 7b)"; however, earlier in the same paragraph, the authors state that DAPT treatment was initiated after engraftment. Clarification is needed as to how DAPT is negatively affecting CLL cell numbers. More importantly with respect to the model, is there any evidence that DAPT treatment had an effect on Wnt signaling in CLL cells in the model?

18. The staining for beta-catenin in the CLL primary cells in tissues shown by the authors is not completely convincing. The authors point to similar staining in myeloma cells, but other authors have reported other distributions for beta-catenin staining in myeloma cells (e.g., Sukhdeo K, et al. *Leukemia* 2012; 26; 116-9). More directly to the matter at hand, while LEF1 expression in CLL is virtually universal, making it a useful diagnostic marker, others have reported that nuclear beta-catenin staining is usually absent in human CLL cells in tissue biopsies (e.g., Tandon et al. *Modern Pathol* 2011; 24; 1433-1443), whereas perinuclear staining is often observed. The onus is on the authors to provide convincing evidence for nuclear beta-catenin localization in primary tumor cells, since the published literature does not support this claim and shows different predominant patterns of beta-catenin localization than that reported by the authors.

Response to Reviewer 1:

2. The Authors excluded a role for Notch receptors other than Notch2, demonstrating a reduction of Notch1 activation upon co-culture and poor expression of Notch3 and Notch4; the latter finding is inconsistent with previous papers clearly showing, through flow cytometry and western blot, that Notch 3 and Notch 4 are highly expressed by MSCs. The Authors should implement Western blot analysis by using anti-Notch3 and anti-Notch-4 antibodies capable of recognizing all the forms of each receptors, including cleaved (active) and full receptors.

As the reviewer points out correctly, Notch3 and Notch4 expression have been described on human BMSCs at higher levels (Nwabo Kamdje *et al.*, 2012) than we observed on EL08-1D2 cells. We believe the most likely explanation for this difference is the use of different cell types or different culture conditions. We have now specified that our data refer to EL08-1D2 cells in the result section on page 5.

Following this reviewer's suggestion, we have added Western blot analyses for Notch3 and Notch4 receptors (Supplement Figure 2f) – we have not observed a CLL-induced cleavage of Notch3 or Notch4. Notably, Notch3 and Notch4 lack a transactivation domain and clearly do not compensate for the loss of Notch2 in our system.

Figure 2: the Authors show Notch expression in CLL samples and stromal cells, but Western Blot and flow cytometry are presented alternatively. The two techniques are complementary and should be used together for each set of experiments. In addition, to validate the expression of ICD1 and ICD2, the Authors should show the results of the analysis following cell exposure to a specific Notch inhibitor, such as DAPT.

As requested by this reviewer, we have now completed our dataset, showing Notch ligand and receptor expression on CLL cells and stromal cells respectively, by flow cytometry and western blot (Figure 2 and Supplement Figure 2). In addition, we have included IF for Notch2 on EL08-1D2 cells in the absence or presence of a Notch inhibitor (Figure 2e and Appendix Figure 1 below). These data show that DAPT inhibits the CLL-induced up-regulation of Notch2.

Appendix Figure 1: EL08-1D2 cells were cultured in the presence or absence of CLL cells. GSI were added as indicated. After 48 hours cells were fixed and stained for Notch2. All images were acquired using identical exposure time.

3. The Authors reported the initial observations using co-culture of 6 CLL samples on EL08-1D2. Regardless the demonstration of the supportive effects of EL08-1D2, human instead of murine MSCs should be preferably used to assess the interactions between leukemic and stromal cells, at least for this introductory experiment. Accordingly, in Figure 1, cocultures should be carried out with human MSCs to confirm the findings obtained with EL08-1D2.

Following this reviewer's suggestion, we have now included more data in our manuscript, confirming that human BMSCs behave no differently to murine BMSCs or EL08-1D2 cells with regard to results from our experiments. These data show that: 1. human BMSCs support the survival of primary CLL cells in a contact dependent manner (Supplement figure 1e). 2. Notch2 is induced in hBMSCs by CLL cells (Figure 2e and

Supplement Figure 2e) and 3. hBMSCs induce the expression of beta-catenin and of N-cadherin in CLL cells (Figure 4b and Supplement Figure 5a, respectively). 4. The Wnt-inhibitor XAV939 mitigates CLL survival-signals from human BMSCs (Supplement Figure 6b).

Classic culture media, such as complete RPMI, are not adequate to support efficiently leukemic cell growth. The difference in expression between co-cultured cells and cells cultured alone could simply reflect a culture artefact, which could be prevented by using alternative culture media.

We have tested how different culture media affect the survival of primary CLL cells and compared this to our standard culture conditions (RPMI). The data, provided here for the reviewer's benefit (Appendix Figure 2), show that, although there are differences in the degree of spontaneous apoptosis, different culture media do not perform better than RPMI or compensate for the survival benefit mediated by cell-to-cell contact between CLL cells and mesenchymal cells.

Appendix Figure 2: Primary CLL cells from 4 patients were cultured for 48 hours in the absence or presence of EL08-1D2 cells. Different culture media were used for mono-cultured cells as indicated (all were supplemented with 10% FCS, last bar indicate RPMI supplemented with 20% FCS). % of live cells was assessed by staining cells for Annexin V/ DAPI.

4. The Authors stated that C1q addition rescued Notch2 deficiency; this concept should be explained better. DAPT did not seem to affect the viability of CLL cells if cultured alone. Nwabo et al. (BCJ 2012) described CLL cell apoptosis by adding gamma-secretase inhibitors when cells were grown both alone and in co-culture. Similarly, Gandhirajan et al. (Neoplasia 2010) demonstrated that CLL cells cultured alone underwent apoptosis when treated with small molecule inhibitors of the Wnt/ β -catenin pathway. The Authors should try to explain these different findings. MSCs physiologically express high levels of β -catenin that is necessary for their anti-apoptotic functions. Thus, one cannot exclude that Wnt/ β -catenin pathway inhibitors act indirectly by modulating the supportive properties of MSCs.

We apologise that our wording may have left unclear what we were referring to; we have now rephrased the sentence to "...C1q addition compensated for Notch2 deficiency of stromal cells and inhibited GSK- β mediated degradation of β -catenin ..." (page 8).

Nwabo et al. described direct cytotoxic effects of GSI XII at concentrations above 5 μ M (Figure 1A in their paper), however, we had not observed pro-apoptotic effects with DAPT at a concentration of 10 μ M in mono-cultured CLL cells. Most likely this discrepancy is related to the different inhibitors used and differences in the *in vitro* bioavailability/ potency of each inhibitor.

In the Gandhirajan paper, direct cytotoxicity on CLL cells is described for CGP049090 and PKF115-584 (at 5 μ M- their Figure 4). We had not observed pro-apoptotic effects with XAV939 and Dvl-PDZ3, but ICG-001 also caused apoptosis in mono-cultured cells similar to the inhibitors used in the Gandhirajan paper (our previous figure 6e).

CGP04090 and PKF115-584 both disrupt the interaction with beta-catenin and TCF, indicating a similar mechanism of action as ICG-001. As XAV939 and Dvl-PDZ-inhibitors operate further upstream in the Wnt-signalling pathways, the lack of direct cytotoxic effects may be related to a different mode of action. In addition, non-specific effects always remain a concern with such experiments.

Importantly, following reviewer 3 suggestion, we have now removed the mono-culture data from this figure as they were not crucial for our manuscript.

Lastly, we appreciate this reviewer's comment that we cannot fully exclude that Wnt-inhibitors act indirectly on CLL cells by affecting MSCs. This aspect of the experiment is now discussed in the result section (page 11). Notably, we had not observed a direct cytotoxic effect of Wnt-inhibitors on MSCs (Supplement Figure 6a). To further address this reviewer's question, we have performed an additional experiment, in which we had pre-treated hBMSCs with the Wnt-inhibitor XAV939 at a dose of 50 μ M for 24 hours (or vehicle control). XAV939 was then removed before culturing CLL cells on these pre-treated MSCs. Results from this experiment (depicted below in Appendix Figure 3) show that inhibition of constitutive active Wnt-signalling in MSCs has no effect on the survival of CLL cells (as opposed to the effect in direct co-culture; Figure 6f). This further supports that Wnt-inhibitors act predominantly directly on stroma-activated CLL cells (although we cannot fully exclude additional effects on CLL-activated stromal cells with the experiment shown in appendix figure 3).

Appendix Figure 3: hBMSCs in mono-culture were treated for 24 hours with the Wnt-inhibitor XAV939 (50 μ M) or vehicle control. Subsequently the inhibitor was removed by repeated washing before primary CLL cells were cultured on these pre-treated cells. Apoptotic CLL cells were analysed 48 hours later by staining for AnnexinV/DAPI (n=6).

Response to Reviewer 2:

1. A more complete description either in the methods or the results section of the transwell assay shown in Figure 1e. No mention of the technical details of this experiment is included in the methods. For example, after transwell cultures, what population of cells were analyzed for apoptosis? CLL cells only, or were they mixed with BSMCs? When CLL cells are co-cultured with BSMCs followed by apoptosis analysis, how is CLL-specific apoptosis identified? Is there staining for CD19 and Annexin V? (Supp Fig 5a does not seem to suggest this). Or are the cells purified by CD19 selection prior to flow analysis?

We are sorry to not have given more details in the methods section regarding this experiment. Firstly, only PBMCs from highly leukemic patients were used for our experiment. Cells were further purged by using anti-CD2 and -CD14 beads to yield >95% purity of CLL cells before these cells were used for further experiments. After co-culture on MSCs, CLL cells can be separated from the co-culture by repeated washing with PBS, allowing the removal of >98% of CLL cells (*Lutzny et al., 2013*). Separated CLL cells are analysed for apoptosis by flow cytometry and staining for Annexin V/ DAPI. As we are using only highly enriched PBMSCs from CLL patients, AnnexinV/ DAPI staining is specific for CLL cells. We have now added an additional panel in Supplement Figure 8a (co-cultured CLL cells) demonstrating this. In addition, MSCs can easily be distinguished from CLL cells by FCS and SSC and gated out for further analyses. For the transwell experiments, primary CLL cells were never mixed with BSMCs and apoptosis assay was performed on CLL cells only. This information has now been added to the supplemental material and method section.

2. In Figure 3f, it should be noted in the legend why several genes are highlighted in red (soluble factors?).

We apologise that we have missed to explain why some genes were marked in red in our previous version of the manuscript. Genes listed in red have known functions in Wnt-signalling. This has now been spelled out in the figure legend.

3. In Figures 1b and 3d, there are two different y-axis – yet these are not described in the legend. Which axis refers to the histogram, and which axis refers to the line graph?

We thank the reviewer for pointing out that this was not made clear in our first submission of the manuscript. We have now added this information to the figure legend and also colour-coded the two axis.

4. The authors show that N-cadherin expression in CLL cells is increased transcriptionally when in co-culture with BSMCs, and that this upregulation is dependent on N-cadherin+ BSMCs and is coincident with beta-catenin stabilization. Is the N-cadherin gene a direct target of TCF/LEF/B-cat complex? Or is this effect secondary to WNT pathway activation? Furthermore, does pharmacologic WNT pathway inhibition affect N-cadherin expression?

This is an interesting question raised by this reviewer. We are not aware of published data showing that the N-cadherin gene is a direct target of the Wnt-pathway and therefore assume the up-regulation is secondary to Wnt-activation.

To address the last question raised by this reviewer, we have analysed the expression of N-cadherin mRNA expression in CLL cells cultured on stromal cells in the presence or absence of the Wnt-inhibitor ICG-001. These data, now shown in Supplement Figure 5g indicate that inhibition of the Wnt-pathway leads to a secondary down-regulation of N-cadherin.

5. *It is well-understood that Notch1 pathway activation is observed in a subset of CLL cases, either by mutation of the NOTCH1 gene or through other mechanisms. Thus, the activity of Notch pathway antagonists such as gamma-secretase inhibitors (eg DAPT) on CLL cell survival is likely a combination of effects on BSMCs and CLL cells themselves. The authors are sure to mention this possibility in their Discussion, yet it would be informative to know what the contribution of BSMCs are to Notch signaling in CLL cells. Co-culture does not induce Notch1 cleavage/activation in BSMCs, but what about in CLL cells? While an in-depth study of the relative effects of DAPT on CLL versus BSMCs in vitro or in vivo is outside the scope of this study, it will nevertheless be very important to begin to understand this if it is to be pursued as a therapeutic strategy as the authors suggest it should. So, for example, in Figure 6d – of the 11 patient CLL samples used for the DAPT treatment study, how many are Notch1 ICD positive? Is there a difference in DAPT effects between Notch1 ICD+ and ICD- samples? If yes, this suggests dual effects on BSMCs and CLL cells are contributing to apoptosis. If no, this may suggest that gamma-secretase inhibition could be acting primarily through the novel mechanism described in this manuscript*

We appreciate this comment and fully acknowledge that stromal-mediated activation of Notch signalling in CLL is important to understand the relative contribution of Notch-inhibition in either cell compartment (stromal cells & CLL cells) for the impaired engraftment of the disease in transplanted NSG- and DAPT treated mice. Jitschin et al. reported that stromal contact activates Notch-signalling in CLL cells, accompanied by increased Notch1 surface expression (*Jitschin et al., 2015*). In order to address this reviewers question whether DAPT is equally effective in NOTCH1 mutated and un-mutated CLL, we sequenced 25 samples for the presence of NOTCH1 mutations. Unfortunately, none of our bio-banked samples carried NOTCH1 mutations. This is consistent with the low frequency of NOTCH mutations in untreated patients (*Puente et al., 2011; Wang et al., 2011*). In order to properly address this question, we estimate we would need to compare at least 10 NOTCH1 mutated and 10 un-mutated CLL cases, for which we would need to sequence at least 100 different primary CLLs. Notably, we have observed pro-apoptotic effects of DAPT in the majority of CLL cells tested, arguing that DAPT certainly affects the viability of NOTCH1 wild-type CLL. Therefore, as discussed, we believe DAPT effects are multifactorial and based on NOTCH-inhibition in CLL *and* stromal cells. As this reviewer points out, this question is outside the scope of our manuscript and we regret that we are not able to address his/her specific question because of the lack of sufficient numbers of NOTCH1 mutated CLLs available to us.

Based on a recent publication from the Dalla-Favera lab, this issue appears to be even more complex, as they have described NOTCH activation in 50% of peripheral blood derived CLL cells in the absence of activating mutations on the NOTCH1 gene (*Fabbri et al., 2017*). Unfortunately, they were unable to show how NOTCH is being activated in these cells. In order to address the relative contribution of Notch inhibition (on cell viability) in each compartment, we had deleted Notch2 from stromal cells (Figure 6a), clearly showing that stromal Notch-activation contributes to the protective effect of stromal cells on CLL cells.

Response to Reviewer 3:

1. The authors' focus on CLL-to-stroma Notch signaling runs counter to the prevailing model in the field of stroma-to-CLL Notch signaling as a critical oncogenic mechanism in CLL, which is supported by significant prior literature, and by the strong genetic signature of selection for activating Notch receptor gene mutations in CLL and related small B cell lymphomas. Stroma-to-lymphocyte Notch signaling is also known to play an important role in sustaining non-neoplastic lymphocyte populations, such as thymic T cell progenitors, marginal zone B cells, and T-follicular helper cells. There are well described barriers that inhibit a given cell from simultaneously acting as a Notch ligand-presenting "sending" cell and Notch receptor-signal "receiving" cell, most notably the phenomenon of Notch receptor-ligand cis-inhibition. One can imagine ways in which this barrier might be overcome, but the authors do not discuss or address this issue at all.

We fully agree with this reviewer that there is strong evidence that Notch-signalling in CLL is a critical oncogenic event. Such activation of Notch can be based on mutations in the Notch-gene or, alternatively, through contact to MSCs (Jitschin et al., 2015; Nwabo Kamdje et al., 2012). However, we do not believe that our data "run counter to the prevailing model in the field of stroma-to-CLL Notch signalling" as we describe a new signalling pathway from CLL-to-stroma. We appreciate the reviewer's suggestion to discuss Notch receptor-ligand cis-inhibition. As Notch cis-inhibition appears to be regulated by the amount of Notch-ligands (e.g. Delta) and Notch-receptors in a given cell, small changes in the expression of either protein can possibly switch a cell from receiving to sending (Sprinzak et al., 2010). As our data clearly show that Notch2 is activated by CLL cells, we must assume that the absolute expression of Notch2 on MSCs exceeds the absolute amount of free ligand expressed in cis. This is now discussed accordingly in the discussion section. Notably, CLL cells also express Notch receptors and ligands and Notch signalling can still be activated by MSCs (Jitschin et al., 2015; Nwabo Kamdje et al., 2012), indicating that cis-inhibition is not an absolute determinant in this cell-to-cell interaction.

2. Fig. 1. Important details of the co-culture assays are not stated in the methods, including whether experiments were done with freshly isolated or cryopreserved cells, and the total number (not just concentration) of cells added to a given well size.

We apologize for this shortcoming. We have now added this information in the material and method section. Importantly, we have not observed differences between cryopreserved and freshly isolated cells with respect to the constitutive expression of beta-catenin (please see appendix figure 4) or N-cadherin.

Appendix Figure 4: β -catenin expression in freshly isolated or cryopreserved primary CLL cells from 3 patients; for a control, CLL cells were co-cultured for 24 hours on EL08-1D2 cells.

3. Fig. 1. The results of the RNA-seq experiments presented in Fig. 1 are difficult to interpret, since the authors compare cells in low-density monoculture for 4 hr (unclear if this is after fresh isolation or after thawing) to cells that have been cultured for 48 hr in the presence of stromal cells. Differences in transcript levels could be due to any of a number of variables that are different between these two conditions (stromal contact, CLL-CLL contact, time of exposure to FCS and other media components etc). These multiple variables undoubtedly all contribute to the large number of genes whose expression is altered in cells co-cultured with EL08-1D2 cells relative to control cells. What about RNA-seq on uncultured primary cells? In other words, to what extent are stromal cells changing gene expression versus maintaining gene expression that exists prior to culture? The same question pertains to the proteomic experiments described in figure 1. The finding in figure 1E that direct contact with stromal cells supports survival of CLL cells in low-density culture has been shown repeatedly in the prior literature and is not novel. These concerns make inclusion of these data, at least within anything other than supplementary data, questionable.

The various factors listed by this reviewer could certainly all contribute to the transcriptional changes described in our manuscript. The RNAseq experiments were performed on thawed cells (we apologize that this was not made clear in the previous version of the manuscript). We had chosen a 4-hour time-point as we wanted to avoid transcriptional changes due to cell death in mono-culture. Furthermore, we expect relatively little changes in RNA expression in this short culture period compared to uncultured cells. We also fully acknowledge that this experiment (as every *in vitro* experiment) has limitations, as it cannot fully capture the *in vivo* complexity of (multiple) cell-cell interactions. However, co-culture experiments can *model* important aspects of these interactions and provide useful information. Importantly, several of the activated pathways identified in our experiment were confirmed in other studies using less sensitive methods (Ding et al., 2010; Edelmann et al., 2008; Jitschin et al., 2015; Nwabo Kamdje et al., 2012; Vangapandu et al., 2017).

In order to underscore that our experiment models aspects of *in vivo* cell-cell interactions, we also compared our gene expression data to the gene expression data obtained from fresh biopsies and published by Herishanu and Wiestner (Herishanu et al., 2011);(GSE21029; comparing peripheral blood to bone marrow; $p < 0.05$). Notably, 1054 genes out of 3402 genes (=31%) from this analysis were also significantly regulated in our analysis. When we further analysed all concordantly regulated genes from this analysis with *GeneTrail*, genes were enriched in the following gene sets: Purin Metabolism; $p = 1.37E-002$; Metabolism of lipids and lipoproteins; $p = 1.52E-007$; PI3K/AKT activation; $p = 4.02E-003$; Glucose metabolism; $p = 2.80E-002$; Regulation of actin cytoskeleton; $p = 6.99E-006$. These gene sets match our results depicted in figure 1b. This analysis demonstrates that our *in vitro* experiment can indeed recapitulate *in vivo* cell-cell interactions.

With regard to figure 1E: We agree that this has been shown previously by other groups. However, we prefer to keep this panel in the main figure, as it is important to introduce direct cell-cell contact to the reader. Should this reviewer disagree with this, we are happy to move the panel into the supplement figure section.

4. Figure 2A. Santa Cruz BT antibodies are notoriously unreliable (unlike Cell Signaling antibodies, which are trustworthy). For example, the JAG2 antibody used (H-143), if one looks at the SCBT website, appears to be cross-reacting with a band of ~200 kb in GM-CSF stimulated Jurkat cells, whereas the predicted size of full-length JAG2 is 133 kD (in fact, the SCBT website shows another JAG2 "specific" antibody that is detecting a protein of a completely different size in extracts prepared from a different cell type). In the western blot that is shown, no molecular masses are given, and there appear to be multiple cross-

reactive polypeptides. The onus is on the authors to demonstrate that the bands shown on Western blot correspond to the proteins of interest.

We appreciate this reviewer's comment. We considered post-translational modifications as another possibility for the multiple bands of JAG2, as it is a target of E3 ligases and can be ubiquitinated. This ubiquitination was shown to be required to activate Notch signalling in MSCs induced by myeloma cells (*Takeuchi et al., 2005*). We therefore assume that the multiple bands of Notch ligands, also described in other papers, may reflect post-translational modifications of JAG2.

As we acknowledge the reviewer's concern and we cannot fully exclude that the antibody cross-reacts with other proteins, we have now performed flow cytometry on CLL cells with different antibodies for Notch ligands. These data indicate that the Notch ligands DLL1, JAG1 and JAG2 are indeed expressed on the surface of CLL cells, in agreement with published data (*Nwabo Kamdje et al., 2012*). The western blots have now been labelled (kd) and moved to the supplement data for the benefit of the new data in the main figure.

5. Figure 2B-C. These figures show that NOTCH2 receptor expression on the authors' stromal cell line of choice, EL08-1D2, is much higher (~10-fold) than in bone marrow stromal populations in vivo (this is made less noticeable by the authors' use of a biexponential scale for Notch receptor flow in EL08-1D2, but not in the flow done in vivo populations). This suggests that authors' co-culture model may facilitate stromal Notch2-dependent effects that may not be relevant in vivo.

We regret that we had unintentionally used different scales for the flow cytometry data depicted in figure 2. The scales have now been harmonised. We fully agree with the reviewer that this is an important issue. The baseline expression of Notch2 on murine BMSCs *in vivo* is lower than the expression in cultured cells.

Notably, the expression of Notch2 on EL08-1D2 cells is similar to levels on human BMSCs (please see appendix figure 5).

Appendix Figure 5: Baseline expression of Notch2 was assessed in EL08-1D2 cells and in human BMSCs using flow cytometry. Isotype controls are shown in red.

We have now performed an additional experiment, analysing Notch2 expression *in vivo* after adoptively transferring TCL-1 tumours into C57BL/6 mice. These data show that – similar to our *in vitro* experiments (Figure 2 and Supplement Figure 2)- CLL cells induce an up-regulation of Notch2 receptors on murine BMSCs *in vivo* (please see appendix figure 6 below). Furthermore, DAPT treatment down-regulated Notch2 expression on MSCs *in vivo*, similar to our *in vitro* results (Figure 2f). Therefore, we are confident that our *in vitro* model reliably recapitulates effects of CLL cell on MSCs *in vivo*.

Appendix Figure 6: Notch2 expression analysis in bone marrow stromal Sca1⁺ cells and bone marrow endothelial CD31⁺ cells of mice transplanted with 40*10⁶ TCL1⁺ leukemic cells and treated with 30 mg/Kg DAPT or vehicle control 4 weeks later. DAPT was administered on 4 consecutive days. The gating for MSCs from mouse bone marrow cells is depicted on the left panels. Notch2 expression on BMSCs *in vivo* of transplanted and treated mice is shown on the right as indicated. An additional, independent experiment showed similar results.

6. Figure 2D-G. A) The decrease in “NICD1” levels that occur in EL08-1D2 cells is misleading, since the antibody used in this blot will detect the Notch1 NTM and is not NICD1 specific. B) The identification of bands on the Western blots labeled NICD2 is presumptuous, since the antibody employed in these blots recognizes all Notch2 polypeptides containing the intracellular domain, including the furin processed NTM subunit. Additional proof is needed that NICD2 levels rise in co-cultured EL08-1D2, such as IF for Notch2 or subcellular fractionation.

We appreciate the reviewer’s comment and have now re-labelled the immunoblots shown in figure 2 and supplement figure 2. As suggested by this reviewer, we have also confirmed Notch2 up-regulation and nuclear expression in stromal cells by IF. These data have now been added to the manuscript and can be found in Figure 2e and Supplement Figure 2g).

7. Suppl Table 1 / Suppl Fig. 2B: If CLL cells are in fact activating Notch signaling in the stromal cells via stromal NOTCH2, we would expect to see increased expression of canonical genes that have been identified as direct Notch targets across many tissues, such as NRARP, increased in the Notch2 proficient versus Notch2 deficient (CLL activated) comparison, but these do not appear in the list of 2-fold upregulated genes. Hes1 does appear in Fig. S2B, but no scale is provided and it’s unclear if there is a large or significant increase. Does a western blot for Hes1 in CLL-activated NOTCH2 proficient vs NOTCH2 deficient stromal cells show a significant increase in the former (i.e. can Fig. 6C be replicated with stroma +/- NOTCH2 knockout rather than GSI)?

The heatmap in supplement figure S2B was included to demonstrate that canonical Notch-targets are regulated in CLL-activated stromal cells. Unfortunately, GSEA heatmaps do not provide a colour scale for their analyses. As explained on the website of the Broad Institute (<http://software.broadinstitute.org>) “In a heat map, expression values are represented as colors, where the range of colors (red, pink, light blue, dark blue) shows the range of expression values (high, moderate, low, lowest)”. We have now included a legend into the Supplement Figures according to this statement.

To address this reviewer’s concerns, we now provide the Fluorescence Intensities from the HES1 probe-set in our Microarray data (Appendix figure 7a). These data show that HES1 is induced in MSCs by CLL cells in a Notch2-dependent manner. In addition, we have validated the regulation of HES1 by Notch2 in EL08-1D2 cells using qPCR (Appendix Figure 7b). These data confirm that EL08-1D2 cells do not behave differently from mouse derived BMSCs.

Appendix Figure 7: a) The Fluorescence intensity for the HES1 probe-set in the microarray data (corresponding to Figure 3) are depicted. b) qPCR was performed for HES1 in wild-type or Notch2-deleted EL08-1D2 cells, co-cultured with CLL cells (n=5).

We are confident that CLL cells are in fact activating Notch2 in MSC based on: 1. Western blot analyses (Figure 2d&e, Supplement Figure 2e), 2. Immunofluorescence, showing nuclear expression of Notch2 in CLL-activated stromal cells (Figure 2f) and 3. Gene expression profiles (Figure 3).

Importantly, our data identify complement C1q as Notch2-regulated genes, important for the activation of Wnt in CLL cells. A contribution of other, canonical Notch-genes to Wnt activation in malignant B cells is outside the focus of our manuscript.

8. What, if anything, is known about the genotype of the CLL cells that were used to stimulate stromal cells? We are told in some experiments that they are not Notch mutated, but if anything else is known about the molecular-cytogenetic character of these samples, it should be provided for completeness.

As suggested by this reviewer, we have now added clinical information to the samples used for our experiments. A new table (Supplement Table 3) is added to the manuscript, providing information on patient's Binet stage, number of previous therapies, cytogenetics, mutational status and degree of dependency on stromal Notch2 for beta-catenin stabilisation.

9. Fig. 3. Notch2 has an important role in regulating the differentiation of certain bone marrow stromal cells, and consistent with this there are differences in gene expression between the Notch2 floxed and Notch2^{-/-} BMSCs. What are the genes/pathways that define these differences? Conversely, CLL cells clearly have large effects on Notch2^{-/-} BMSCs, based on PCA. What are the genes/pathways that define these differences? Arguably, these uncontrolled variables make it very difficult to infer precisely what the role of Notch2 is in this system. Other approaches, such as use of antibodies that specifically block Notch2 signaling, might yield more easily interpretable results and at a minimum would provide complementary information.

We have now added a pathway analysis of the genes, which are constitutively regulated in MSCs by Notch2 in the absence of CLL cells, corresponding to the light blue and orange gene clusters depicted in Figure 3c (Supplement Figure 3a).

We agree with this reviewer that CLL cells have large effects on the gene expression in MSCs, irrespectively of Notch2. However, we disagree with this reviewer that those effects are uncontrolled as they are present in Notch2 proficient and Notch2 deficient MSCs (and are therefore controlled for). The only variable in the experiment is the

expression of Notch2 in MSCs, therefore the experiment addresses exactly the role of Notch2 in CLL activated MSCs. Blocking Notch2 signalling with a blocking antibody would not only affect Notch2 signalling in MSCs, but also Notch2 signalling in CLL cells (as CLL cells express also Notch2 receptors (Rosati et al., 2009)). Therefore, such an experiment would not truly complement our initial experiment and results would not specifically reflect the role of Notch2 in MSCs.

10. A potentially intriguing observation is the upregulation of beta-catenin in CLL cells by co-culture with Notch2-replete BMSCs and not by Notch2-/- BMSCs. The experiments in Fig. 4A-B are not adequately described in the legends. If the comparison is between CLL cells kept in low-density culture for 5 days +/- stromal cells, the CLL in the no-stroma group may be largely apoptotic (see Fig. 1E); presumably this is not the case, but it needs to be spelled out. The 24 hr conditioned medium lane in Fig. 4C shows some evidence of beta-catenin upregulation by conditioned media (as the authors discuss later); the size of the effect should be clarified by providing a better Western blot, and the effect should be replicated across multiple CLL samples. Again, complementary experiments with acute perturbagens (GSI, blocking N2 antibody) would help to build confidence in a direct link between Notch2 and beta-catenin stabilization. The experiments with CRISPR/Cas9 Notch2 knockouts are helpful to mitigate this concern, but the methods do not explain how these cells were isolated (single cell clones, pooled clones, etc.). Additional information is needed.

We apologize for the shortcoming of not providing more detailed information for the experiments depicted in figure 4A and 4B and how CRISPR knock-out cell have been generated. In this experiment CLL cells were cultured for 5 days- viability was checked at the end of the culture to ensure that cells were not largely apoptotic (notably, a 24 hour time point gave similar results; please see figure 4g). In addition, as shown in the time course experiment in figure 5a, beta-catenin induction in CLL cells by MSCs occurs as early as 12 hours, at a time point when apoptosis in mono-culture can be neglected. The experiments depicted in figure 4c had of course been replicated across several different primary CLL samples. The results were identical and the western blot as shown represents this finding. We now provide the entire data-set from repeated experiments for this reviewer's benefit (Appendix Figure 8).

Appendix Figure 8: β -catenin expression was evaluated in primary CLL cells after 24 and 48 hours in direct co-culture with EL08-1D2 (lanes 1+2). Lane 3: Conditioned media (CM) from the 48 hours co-cultures or fresh medium (lane 4) was used as culture-medium for freshly thawed cells of the same patient. β -catenin expression was assessed after 24 hours.

CRISPR knock-out cells have been generated by knocking out Notch2 and puromycin-selection in primary bulk cells derived from the bone marrow of wild-type mice. We had not selected single clones to avoid the selection of sub-clones with potential (though unlikely) other genetic lesions. We have included more information into the supplemental material and methods sections to clarify this.

As suggested by this reviewer, we have now included data into the manuscript showing that DAPT treatment down-regulates beta-catenin in co-cultured CLL cells (Figure 6e).

11. Fig. 4. The authors do not show whether the CLL beta-catenin induction effect is also mediated by mBMSC's +/- Notch2 deletion (only +/- EL08-1D2 co-culture data is shown). It is important to show that the stromal Notch2 / CLL beta catenin effect is not an idiosyncratic property of the EL08-1D2 cell line, which seems to have unusually high expression of NOTCH2.

We agree that this would be a relevant concern. However, we had already shown primary mBMSCs depleted of Notch2 by CRE recombinase in our first submission of the manuscript (Figure 4e). Therefore, Notch2 regulation of beta-catenin is not an idiosyncratic property of EL08-1D2 cells.

12. Fig. 5A-E and I-J. These experiments establish that co-culture facilitates CLL N-cadherin expression, and that this effect and CLL beta-catenin expression are dependent on stromal N-cadherin and (in the co-culture system) on stromal Notch2. The authors should show that this effect is not simply due to increased CLL-CLL contacts facilitated by aggregation on stromal cells, by comparing stromal co-culture to high-density CLL monoculture, or culture of CLL cells in round-bottom plates that force aggregation of CLL cells.

To address this reviewer's concern, we have performed experiments with different cell densities and with round-bottom plates. Results from this experiment are depicted in appendix figure 9 and indicate that the cell density of CLL cells does not affect N-cadherin/ beta-catenin expression in CLL cells.

Appendix Figure 9: N-cadherin expression was evaluated in primary CLL cells after 24 hours of co-culture with EL08-1D2, high cell density monoculture or monoculture in round well plates. The number of CLL cells in 1ml medium is indicated.

13. Fig. 5F-H. These are interesting results, but they are confined to the EL08-1D2 cell line, which as discussed above, may not be representative of stromal cells that CLL cells are likely to encounter in vivo.

We agree with the reviewer that EL08-1D2 cells may have unique properties, which may limit the extrapolation of conclusions drawn from data obtained from these cells to other cells. However, we have not found any difference between EL08-1D2 cells, primary mouse MSCs and primary human BMSCs with regard to N-cadherin and beta-catenin expression in CLL cells. We therefore believe that EL08-1D2 cells are useful tools to study CLL cell-MSK interactions.

14. Fig 6A. The prosurvival effect in CLL cells that is dependent on stromal NOTCH2 appears to be quite modest. Furthermore, the authors seem to have combined data from two different experiments (co-culture with cre-deleted Notch2 and Cas9-deleted Notch2 in MSC's) in order to make the figure, raising the question of whether either of the two original experiments on their own showed a significant effect.

To address this reviewer's question, we have analysed the effects of Notch2 deletion by Cas9 and by CRE recombinase in separate analyses. The data are depicted for this reviewer's benefit (Appendix Figure 10) and show no difference in the relative change of viability between the different experiments.

Appendix Figure 10: The graphs show the % of apoptotic primary CLL cells following 5 days co-culture on cre-deleted Notch2 (left) and Cas9 deleted Notch2 (right) bone marrow stromal cells.

Could the difference in CLL survival on Notch2+/- stroma be more impressive for that subset of CLL samples with a strong stromal NOTCH2 dependency for beta-catenin expression (per figure suppl 3e), or is there no correlation?

As suggested by this reviewer, we have now analysed whether there is a correlation between the dependency on stromal Notch2 (for β-catenin expression in CLL) and CLL cell survival on Notch2 deficient stroma (compared to Notch2 proficient stroma). The data are depicted here in Appendix Figure 11 and show that there is a correlation between these two variables.

Appendix Figure 11: CLL cell viability on Notch2 proficient or deficient stroma was assessed as described in figure 6a. In parallel, β-catenin levels in CLL cells (N=10), cultured on Notch pro- or deficient stromal cells, were analysed by western blotting and quantified by using ImageJ© software.

Also, why was this experiment not also conducted with EL08-1D2 stromal cells +/- Notch2 deletion, the primary model used by the authors in Fig. 4 experiments?

As mentioned above, we had introduced primary mBMSCs in figure 4 and observed similar effects in EL08-1D2 cells and mBSCs with regard to all aspects of our work. Therefore, we believe repeating this experiment with EL08-1D2 cells would be redundant and not further advancing our manuscript.

15. Fig. 6D-E. The y-axis should be plotted as the absolute % apoptotic cells rather than apoptosis "relative to untreated cells" since this hides the baseline apoptosis values for the monoculture (which are likely very high). It may be best to simply remove the monoculture data from this figure. Statistical comparisons should be vs. untreated cells in the same culture system – it's unclear what is the biological significance of comparing relative difference in % apoptosis across culture systems with very different baseline levels of apoptosis.

We thank the reviewer for this comment; as the monoculture are indeed not crucial for the figure, these were removed and the data re-analysed accordingly. Cell viability is now indicated with absolute numbers.

16. The DAPT experiments in Fig. 7 are difficult to interpret because this treatment will block Notch signaling in the CLL cells as well as in the EL08-1D2 cells. Clearly, Notch signaling is important in these cells since there is selected pressure for Notch gain of function mutations in CLL. Similarly, the same confounding issue hangs over the in vivo PDX modeling work in which mice are treated with DAPT. To ignore this facet of the system seems an important oversight.

We agree with this reviewer that the experiments with DAPT do not allow to distinguish between effects on stroma and CLL cells (the optimal experiment would be an adoptive transfer of TCL-1 tumour cells into Notch2 deficient mice- however, since the germline deletion of Notch2 leads to embryonic lethality such an experiment cannot be done for technical reasons). We have not "ignored this facet of the system" as this was already discussed in the previous version of the manuscript (previous pages 15/16)

Similarly, the authors never discuss possible Wnt signaling/beta-catenin expression in BMSCs; can effects on stromal cells be excluded in experiments with Wnt inhibitors?

We thank the reviewer for this comment and agree that we cannot fully exclude that Wnt-inhibitors have direct effects on MSC, which may also impinge on the interaction between MSCs and CLL cells. To address this concern, we had excluded that these inhibitors have cytotoxic effects on MSCs (Supplement Figure 6a). In addition, we have now performed an experiment in which we pre-treated stromal cells with the Wnt-inhibitor XAV939 and co-cultured CLL cells after removal of the inhibitor. These data show that inhibition of Wnt-signalling, before co-culturing CLL cells, has no significant effect on the survival of CLL cells (as opposed to the toxic -likely direct- effects of Wnt-inhibitors in co-culture- please see our response to reviewer 1, comment 4, appendix figure 3). As suggested by this reviewer and reviewer 1, we now discuss this issue further in the result section (page 11).

17. In the description of the results in Fig. 7, the authors state that "the degree of infiltration was assessed by staining cells for human-CD45+, CD19+, CD5+ and flow cytometry, confirming that DAPT treatment impaired splenic engraftment in the CLL-PDX model (Figure 7b)"; however, earlier in the same paragraph, the authors state that DAPT

treatment was initiated after engraftment. Clarification is needed as to how DAPT is negatively affecting CLL cell numbers. More importantly with respect to the model, is there any evidence that DAPT treatment had an effect on Wnt signaling in CLL cells in the model?

We apologise that the description of the experiment was confusing. DAPT treatment was initiated 5 days after *transplantation and homing* of cells to lymphoid tissues (homing happens in the first 48 hours). This has now been corrected. In order to further address how DAPT affects engraftment of CLL cells in mice, we have now performed an additional experiment with the TCL-1 mouse model. After transplantation of syngeneic, FarRed CellTracker-labelled tumour cells (from diseased TCL-1 mice) into un-irradiated C57BL/6 wild-type mice, mice were treated with DAPT for 4 consecutive days. Results from this experiment demonstrate that DAPT treatment impairs cell proliferation of malignant B cells. As the pro-apoptotic effect of DAPT *in vitro* is relatively small but the differences in engraftment relatively higher *in vivo*, we assumed that impaired cell proliferation could also contribute to this phenotype in addition to pro-apoptotic effects (this was discussed in our previous draft of the manuscript). These data have now been added as Supplement Figure 7). Furthermore, our new data confirm that DAPT also down-regulates Notch2 in MSCs *in vivo* (please see appendix figure 6), similar as observed in our *in vitro* experiments (Figure 2e), indicating that DAPT blocks Notch signalling in MSCs *in vivo*. Staining for murine beta-catenin in TCL-1+ tumour cells was unfortunately unsuccessful as the antibodies we used showed a non-specific binding.

18. The staining for beta-catenin in the CLL primary cells in tissues shown by the authors is not completely convincing. The authors point to similar staining in myeloma cells, but other authors have reported other distributions for beta-catenin staining in myeloma cells (e.g., Sukhdeo K, et al. *Leukemia* 2012: 26; 116-9). More directly to the matter at hand, while LEF1 expression in CLL is virtually universal, making it a useful diagnostic marker, others have reported that nuclear beta-catenin staining is usually absent in human CLL cells in tissue biopsies (e.g., Tandon et al. *Modern Pathol* 2011: 24; 1433-1443), whereas perinuclear staining is often observed. The onus is on the authors to provide convincing evidence for nuclear beta-catenin localization in primary tumor cells, since the published literature does not support this claim and shows different predominant patterns of beta-catenin localization than that reported by the authors.

We thank the reviewer for pointing out that there is a controversy about the constitutive activation of β -catenin *in vivo*. We and other groups (*El-Gamal et al., 2014; Lu et al., 2004*) have not found evidence for a constitutive activation of β -catenin in peripheral blood (PB) CLL cells, whereas other papers report a constitutive expression of β -catenin in PB CLL cells (*Bojarczuk et al., 2016; Gandhirajan et al., 2010*). The same discrepancies hang over the issue of β -catenin expression in tissues, based on the cited paper and our own data. Notably, our data show that the Notch2-dependent, stromal regulation of β -catenin in CLL cells is extremely robust, as we have observed this in all primary CLL cells tested, with mouse and human derived MSCs. It therefore seems almost impossible to envision that this finding is restricted to *in vitro* conditions. In the Tandon paper, the predominant cytoplasmic expression of β -catenin in CLL cells in the bone marrow seems to contrast our data. We think this is likely related to different antibodies or antigen-retrieval used for the 2 experiments. Our staining of β -catenin is clearly different from the staining observed by Tandon et al.: We find nuclear β -catenin (to different degrees) in a significant subset of cells in the bone marrow and, more frequent, in lymph node sections. We now acknowledge this apparent discrepancy between our data and the Tandon paper by discussing this in more detail in the discussion section of our revised manuscript.

In order to further strengthen our findings from IHC, we have now performed immunofluorescence for β -catenin (Non-phospho (Active) β -Catenin (Ser45)) on lymph node biopsies from CLL patients. Notable, this antibody only recognises activated β -catenin (*Sakanaka, 2002*). These data, depicted here in Appendix Figure 12 unambiguously show that β -catenin is activated in lymph nodes *in vivo* and confirm our results from IHC. (We are currently in the process of optimising this staining also for bone marrow samples; however, as these samples are decalcified, this is not as straight forward as we had hoped).

Appendix Figure 12: Infiltrated Lymph node biopsies from CLL patients were stained with anti-CD19 and anti-beta-catenin (non-phospho serine 45). Secondary antibodies alone were used as control to exclude non-specific binding of secondary antibodies. Scale bars=100 μ m.

In additional support of our proposed regulation of β -catenin by MSCs in other B cell malignancies than CLL, nuclear β -catenin expression was reported in MCL- and DLBCL- lymph-nodes (*Ge et al., 2012; Gelebart et al., 2008*), which may also be driven by signals from the microenvironment.

REFERENCES

- Bojarczuk, K., Sasi, B.K., Gobessi, S., Innocenti, I., Pozzato, G., Laurenti, L., and Efremov, D.G. (2016). BCR signaling inhibitors differ in their ability to overcome Mcl-1-mediated resistance of CLL B cells to ABT-199. *Blood* 127, 3192–3201.
- Ding, W., Knox, T.R., Tschumper, R.C., Wu, W., Schwager, S.M., Boysen, J.C., Jelinek, D.F., and Kay, N.E. (2010). Platelet-derived growth factor (PDGF)-PDGF receptor interaction activates bone marrow-derived mesenchymal stromal cells derived from chronic lymphocytic leukemia: implications for an angiogenic switch. *Blood* 116, 2984–2993.
- Edelmann, J., Klein-Hitpass, L., Carpinteiro, A., Führer, A., Sellmann, L., Stilgenbauer, S., Dührsen, U., and Dürig, J. (2008). Bone marrow fibroblasts induce expression of PI3K/NF-kappaB pathway genes and a pro-angiogenic phenotype in CLL cells. *Leuk. Res.* 32, 1565–1572.
- El-Gamal, D., Williams, K., LaFollette, T.D., Cannon, M., Blachly, J.S., Zhong, Y., Woyach, J.A., Williams, E., Awan, F.T., Jones, J., et al. (2014). PKC- β as a therapeutic target in CLL: PKC inhibitor AEB071 demonstrates preclinical activity in CLL. *Blood* 124, 1481–1491.
- Fabrizi, G., Holmes, A.B., Viganotti, M., Scuoppo, C., Belver, L., Herranz, D., Yan, X.-J., Kieso, Y., Rossi, D., Gaidano, G., et al. (2017). Common nonmutational NOTCH1 activation in

chronic lymphocytic leukemia. *Proc. Natl. Acad. Sci. U.S.a.* 201702564.

Gandhirajan, R.K., Staib, P.A., Minke, K., Gehrke, I., Plickert, G., Schlösser, A., Schmitt, E.K., Hallek, M., and Kreuzer, K.-A. (2010). Small molecule inhibitors of Wnt/beta-catenin/lef-1 signaling induces apoptosis in chronic lymphocytic leukemia cells in vitro and in vivo. *Neoplasia* 12, 326–335.

Ge, X., Lv, X., Feng, L., Liu, X., and Wang, X. (2012). High expression and nuclear localization of β -catenin in diffuse large B-cell lymphoma. *Mol Med Rep* 5, 1433–1437.

Gelebart, P., Anand, M., Armanious, H., Peters, A.C., Dien Bard, J., Amin, H.M., and Lai, R. (2008). Constitutive activation of the Wnt canonical pathway in mantle cell lymphoma. *Blood* 112, 5171–5179.

Herishanu, Y., Pérez-Galán, P., Liu, D., Biancotto, A., Pittaluga, S., Vire, B., Gibellini, F., Njuguna, N., Lee, E., Stennett, L., et al. (2011). The lymph node microenvironment promotes B-cell receptor signaling, NF-kappaB activation, and tumor proliferation in chronic lymphocytic leukemia. *Blood* 117, 563–574.

Jitschin, R., Braun, M., Qorraj, M., Saul, D., Le Blanc, K., Zenz, T., and Mougiakakos, D. (2015). Stromal cell-mediated glycolytic switch in CLL-cells involves Notch-c-Myc signaling. *Blood* 125, 3432–3436.

Lu, D., Zhao, Y., Tawatao, R., Cottam, H.B., Sen, M., Leoni, L.M., Kipps, T.J., Corr, M., and Carson, D.A. (2004). Activation of the Wnt signaling pathway in chronic lymphocytic leukemia. *Proc. Natl. Acad. Sci. U.S.a.* 101, 3118–3123.

Lutzny, G., Kocher, T., Schmidt-Supprian, M., Rudelius, M., Klein-Hitpass, L., Finch, A.J., Dürig, J., Wagner, M., Haferlach, C., Kohlmann, A., et al. (2013). Protein kinase c- β -dependent activation of NF- κ B in stromal cells is indispensable for the survival of chronic lymphocytic leukemia B cells in vivo. *Cancer Cell* 23, 77–92.

Nwabo Kamdje, A.H., Bassi, G., Pacelli, L., Malpeli, G., Amati, E., Nichele, I., Pizzolo, G., and Krampera, M. (2012). Role of stromal cell-mediated Notch signaling in CLL resistance to chemotherapy. *Blood Cancer J* 2, e73.

Puente, X.S., Pinyol, M., Quesada, V., Conde, L., Ordóñez, G.R., Villamor, N., Escaramis, G., Jares, P., Beà, S., González-Díaz, M., et al. (2011). Whole-genome sequencing identifies recurrent mutations in chronic lymphocytic leukaemia. *Nature* 475, 101–105.

Rosati, E., Sabatini, R., Rampino, G., Tabilio, A., Di Ianni, M., Fettucciari, K., Bartoli, A., Coaccioli, S., Screpanti, I., and Marconi, P. (2009). Constitutively activated Notch signaling is involved in survival and apoptosis resistance of B-CLL cells. *Blood* 113, 856–865.

Sakanaka, C. (2002). Phosphorylation and regulation of beta-catenin by casein kinase I epsilon. *J. Biochem.* 132, 697–703.

Sprinzak, D., Lakhonpal, A., Lebon, L., Santat, L.A., Fontes, M.E., Anderson, G.A., Garcia-Ojalvo, J., and Elowitz, M.B. (2010). Cis-interactions between Notch and Delta generate mutually exclusive signalling states. *Nature* 465, 86–90.

Takeuchi, T., Adachi, Y., and Ohtsuki, Y. (2005). Skeletrophin, a novel ubiquitin ligase to the intracellular region of Jagged-2, is aberrantly expressed in multiple myeloma. *Am. J. Pathol.* 166, 1817–1826.

Vangapandu, H.V., Ayres, M.L., Bristow, C.A., Wierda, W.G., Keating, M.J., Balakrishnan, K., Stellrecht, C.M., and Gandhi, V. (2017). The Stromal Microenvironment Modulates Mitochondrial Oxidative Phosphorylation in Chronic Lymphocytic Leukemia Cells. *Neoplasia* 19, 762–771.

Wang, L., Lawrence, M.S., Wan, Y., Stojanov, P., Sougnez, C., Stevenson, K., Werner, L., Sivachenko, A., DeLuca, D.S., Zhang, L., et al. (2011). SF3B1 and other novel cancer genes in chronic lymphocytic leukemia. *N. Engl. J. Med.* 365, 2497–2506.

Reviewers' Comments:

Reviewer #1:

Remarks to the Author:

The Authors clarified all the raised issues and modified the manuscript accordingly, which now supports convincingly that small molecule inhibitors of Notch and Wnt/ β -catenin may have an anti-tumoral role in CLL.

Reviewer #2:

Remarks to the Author:

Mangolini, Götte et al. have substantially improved their manuscript by addressing both my and other reviewer comments and concerns. It is my opinion that the findings described in their study are novel and would be of interest to others in the community. One remaining point concerns resolving the in vivo effects of Notch pathway inhibition, referred to in my original comment #5. The authors make much effort throughout the study to nominate Notch2 pathway activation in MSCs through CLL cell crosstalk as promoting CLL cell survival. This is mechanistically explained through beta-catenin stabilization in CLL cells by a combination of complement-factor signaling and upregulation of N-cadherin. The authors spend much effort citing these in vitro observations as rationale for Notch pathway inhibition as a potential therapeutic strategy for B cell malignancies, and that Notch signaling ablation with gamma-secretase inhibitors could have significant anti-tumor effects not only through direct effects on CLL cells but also indirectly through modulation of Notch2 in MSCs. In their updated manuscript, the authors provide further evidence that this mechanism may be relevant in the in vivo setting, using the adoptive transfer TCL1 model with pharmacodynamic assessment of Notch2 levels (Supp. Figure 7 and Appendix Figure 6). Several aspects to this experiment are unclear: is the antibody used to assess Notch2 detecting ICN (is this experiment reliably assessing activated Notch2 signaling?); if not, can another proxy measurement for active Notch2 signaling in these cells be performed (HES1 expression?); why are these data excluded from the final figures in the manuscript? - it seems to me that in vivo pharmacodynamic evidence of Notch2 pathway perturbation in MSCs in a tumor-bearing animal model significantly bolsters the authors' hypotheses regarding the therapeutic potential and mechanism of action of these agents. Inclusion of these studies and a clarification of the reagents used would further strengthen the study.

Reviewer #3:

Remarks to the Author:

The authors have been mostly responsive to the original critiques. Below lie this reviewer's comments pertaining to the authors' responses to the original critique. Original comments that are not mentioned have been dealt with adequately.

Original comments #1, 6, and 7. In the original critique, the authors were asked to address the issue of cis-inhibition, in which cells expressing an excess of receptor inhibit intrinsic Notch ligand activity, and vice versa in cells expressing an excess of ligand. As mentioned, there are ways one can envision how this could be circumvented, including rapid changes/cycling of relative levels of receptors and ligands in CLL cells and stromal cells. Discussion of this issue in the revised paper is welcome.

The strongest data that pertain to the idea that stromal cells are signal receivers and that this reception involves Notch2 are the genetic data shown in Fig. 3, in which it is evident that deletion of N2 influences CLL-dependent and independent gene expression programs in MSCs (presumably via effects on canonical Notch signaling). The data from experiments using EL08-1D2 cells remain less clear, in part because the corresponding western blots and IF are confusing. The antibodies

the authors used (which are not delineated well in the text or legends and are inferred from the list of reagents in the supplemental methods) are not specific for the active forms of N1 or N2. As a result, the most conservative interpretation of the EL08-1D2 data is that upon incubation with CLL cells the mature Notch1 receptor is downregulated transcriptionally or is degraded, and the mature Notch2 receptor is upregulated transcriptionally or is stabilized. None of the data rigorously prove that there is any change in Notch2 activation in EL08-1D2 cells following exposure to CLL cells.

Original comment #3. Fig. 1. The results of the RNA-seq experiments presented in Fig. 1 to some degree still involve a comparison of apples and oranges. The public data cited by the authors partially mitigate this concern, but also raise additional questions. The major thrust of the Herishanu et al. paper was focused on the importance of the lymph node microenvironment, not the bone marrow microenvironment. Is the signature identified in the co-culture system also identified in the lymph node microenvironment in the Herishanu et al. data sets? This is arguably more important, since most CLL proliferation/growth appears to occur mainly in lymph nodes.

Original comment #4. The authors have not addressed this issue; some of the figures continue to rely on reagents that are likely to be unreliable (e.g., Santa Cruz antibody H-143). The hand waving about ubiquitinylation, etc., is not an adequate response.

Original Comment #9. There is little or no hard evidence suggesting that Notch2 has any role in CLL cells, and the idea of testing a Notch2 blocking antibody remains viable.

Original Comment #18. Of note, although as the authors point out different groups have arrived at different results staining CLL lymph nodes for beta-catenin, the IHC protocol described by Tandon et al. is used widely in cancer diagnostics in clinical pathology departments, which have to be held to a high standard because test results are used to sub-classify cancers and guide patient care. The "staining" shown in Fig. 7C shows nonspecific DAB precipitates and are not worthy of publication in Nature Communications. Furthermore, there is no evidence presented, here or elsewhere, for a role for Notch2 in stromal cells in CLL within human tissues. Absent such connections, the significance of the phenomena described in this paper to human CLL is uncertain.

Response to Reviewer 2:

[.....] In their updated manuscript, the authors provide further evidence that this mechanism may be relevant in the in vivo setting, using the adoptive transfer TCL1 model with pharmacodynamic assessment of Notch2 levels (Supp. Figure 7 and Appendix Figure 6). Several aspects to this experiment are unclear: is the antibody used to assess Notch2 detecting ICN (is this experiment reliably assessing activated Notch2 signaling?); if not, can another proxy measurement for active Notch2 signaling in these cells be performed (HES1 expression?); why are these data excluded from the final figures in the manuscript? - it seems to me that in vivo pharmacodynamic evidence of Notch2 pathway perturbation in MSCs in a tumor-bearing animal model significantly bolsters the authors' hypotheses regarding the therapeutic potential and mechanism of action of these agents. Inclusion of these studies and a clarification of the reagents used would further strengthen the study.

We appreciate this comment and suggestions raised by reviewer 2. Following his/ her advice, we have now included our data using the adoptive transfer Tc1 mouse model into the main figure (figure 7). As -to our best knowledge- no existing FACS antibody allows to discriminate between activated and total Notch2, we followed this reviewer's advice and assessed C1q mRNA transcript levels in MSCs from DAPT treated mice (as a surrogate for Notch2 activation in stromal cells, please see figure 4i). These experiments were technically challenging because they required a rapid harvest of BMSCs from culled mice, staining and flow sorting of Sca1⁺ MSCs for RNA preparations. Because we could initially only recover very low yields of RNA from individual mice, we finally needed to pool cells from several DAPT-treated (or vehicle-treated) mice. Our data, depicted in figure 7, show that DAPT treatment reduces the level of C1q mRNA in MSCs. The effects with DAPT were less pronounced than deletion of Notch2 in MSCs *in vitro* (figure 4i); however, we think this is not unexpected as a pharmacological inhibition rarely leads to a complete functional loss of a protein. Similar to our observations from *in vitro* experiments, DAPT also down-regulated Notch2 on MSCs *in vivo*, suggesting that the dose we chose for treatment was sufficient to target stromal Notch2 *in vivo*.

Response to Reviewer 3:

Original comments #1, 6, and 7. In the original critique, the authors were asked to address the issue of cis-inhibition, in which cells expressing an excess of receptor inhibit intrinsic Notch ligand activity, and vice versa in cells expressing an excess of ligand. As mentioned, there are ways one can envision how this could be circumvented, including rapid changes/cycling of relative levels of receptors and ligands in CLL cells and stromal cells. Discussion of this issue in the revised paper is welcome.

The issue of cis-inhibition is interesting and we acknowledged this by discussing this in our previous submission. We regret if this was not obvious as we had not colour-coded changes in the previous version of our manuscript. On page 17, we wrote: "Furthermore, the expression of Notch-receptors and Notch-ligands on the same cell can lead to cis-inhibition of Notch signalling. Therefore, the transmitted signal strength is also regulated by the stoichiometric relation between ligands and receptors Ref37." We hope the reviewer considers this sufficient, as we feel that a more extensive discussion of this issue is beyond the scope of this manuscript.

The strongest data that pertain to the idea that stromal cells are signal receivers and that this reception involves Notch2 are the genetic data shown in Fig. 3, in which it is evident that deletion of N2 influences CLL-dependent and independent gene expression programs in MSCs (presumably via effects on canonical Notch signaling). The data from experiments using EL08-1D2 cells remain less clear, in part because the corresponding western blots and IF are confusing. The antibodies the authors used (which are not delineated well in the text or legends and are inferred from the list of reagents in the supplemental methods) are not specific for the active forms of N1 or N2. As a result, the most conservative interpretation of the EL08-1D2 data is that upon incubation with CLL cells the mature Notch1 receptor is downregulated transcriptionally or is degraded, and the mature Notch2 receptor is upregulated transcriptionally or is stabilized. None of the data rigorously prove that there is any change in Notch2 activation in EL08-1D2 cells following exposure to CLL cells.

We apologize if it was not clear from our previous version of the manuscript which antibody was used for each experiment. We have now added a table in the supplemental data (material and method section) listing all antibodies used for our studies with reference to the experiment/ figure.

We agree with the reviewer that total Notch2 increases in MSCs activated by CLL cells. To provide more evidence that Notch2 is *activated* in MSCs, we now included data on the nuclear expression of Notch2 in stromal cells: Data from confocal microscopy show an increased abundance of nuclear Notch2 in MSCs co-cultured with CLL cells compared to mono-culture stromal cells. The new data are now included into the main figure 2.

As previously discussed, additional evidences for Notch2 activation in MSCs are based on the following observations and results:

1. CLL contact induces cleavage of Notch2, shown as the cleaved fragment on the western blot depicted in Supplement figure 2.
2. HES1 up-regulation in MSCs upon contact with CLL cells, in a Notch2 dependent manner (Appendix figure 7 of our previous rebuttal letter).
3. Gene expression profiles from MSCs in the presence or absence of Notch2 (Figure 3).

Original comment #3. Fig. 1. The results of the RNA-seq experiments presented in Fig. 1 to some degree still involve a comparison of apples and oranges. The public data cited by the authors partially mitigate this concern, but also raise additional questions. The major thrust of the Herishanu et al. paper was focused on the importance of the lymph node microenvironment, not the bone marrow microenvironment. Is the signature identified in the co-culture system also identified in the lymph node microenvironment in the Herishanu et al. data sets? This is arguably more important, since most CLL proliferation/growth appears to occur mainly in lymph nodes.

Culturing CLL cells on stromal cells *in vitro* is a commonly used way to model bone marrow stroma –CLL interactions. Upon the concerns raised by this reviewer, we had compared our data to data from the Herishanu paper, which are publically available. We compared GEP from this data set from CLL cells in the peripheral blood to those in the bone marrow, as this comes closest to our *in vitro* condition. As acknowledged by this reviewer, we could indeed verify that *ex vivo* can recapitulate *in vivo* conditions with regard to the identified gene sets (please see our previous rebuttal letter, page 7).

We are now asked to compare our data to the gene expression data identified in the lymph node compartment. Although we agree that –for disease progression- CLL proliferation is important, the composition of the bone marrow compartment is very different to the lymph node compartment, where T cell – CLL cell interactions drive cell

proliferation. Stroma cells protect tumour cells from drug-induced and spontaneous apoptosis without inducing proliferation. Therefore, we believe a comparison of our data to the lymph node compartment gene signature described in the Herishanu paper will not further advance our manuscript.

Original comment #4. The authors have not addressed this issue; some of the figures continue to rely on reagents that are likely to be unreliable (e.g., Santa Cruz antibody H-143). The hand waving about ubiquitinylation, etc., is not an adequate response.

We had taken this reviewer's criticism very seriously. We think we had adequately addressed this issue in our previous rebuttal letter: Following this reviewer's concern, we had included an entire new set of data using different flow cytometry antibodies for Notch ligands: These data were presented in Figure 2, confirming with a different method and antibodies that CLL cells express Notch ligands. (The western blot data had been put in the supplement figure).

Our results are in fact a confirmation of published data: The double band for JAG2 on this western blot is entirely consistent with data from Rosati et al.; Blood 2009, PMID 18796623; please see Figure 1 D&E and statement in their result section: "*Jagged1 and Jagged2 analysis shows that B-CLL cells from all 25 patients expressed both ligands, each appearing as a doublet of 150 kDa, whereas these proteins were undetectable in PBMCs, PBLs, and B-CD19 cells from all healthy donors examined (Figure 1D,E)*".

We were unable to find published evidence that this Santa Cruz antibody does not detect Jagged-2.

Additional confirmation of Notch-ligand expression on CLL cells was published by Nwabo Kamdje 2012 (PMID 22829975, table 1).

Original Comment #9. There is little or no hard evidence suggesting that Notch2 has any role in CLL cells, and the idea of testing a Notch2 blocking antibody remains viable.

As discussed in our previous rebuttal letter, published evidence that Notch2 is expressed and activated in CLL cells originates from Rosati et al. (Blood 2009; PMID 18796623). This paper clearly shows constitutive expression and activation of Notch2 in CLL cells (please see their Figures 2 and 4E).

Additional evidence for Notch2 activation in CLL cells can be found in the following papers: Blood 2002; PMID 11986231; "*Notch2 is involved in the overexpression of CD23 in B-cell chronic lymphocytic leukemia*". This paper also shows overexpression and activation of Notch2 in CLL. PMID 19995395 and PMID 15565166 are additional papers describing Notch2 activation in CLL cells.

Therefore, we believe an experiment with a blocking antibody against Notch2 would also directly affect gene expression in CLL cells and is therefore not complementary to our experiment depicted in figure 3.

Original Comment #18. Of note, although as the authors point out different groups have arrived at different results staining CLL lymph nodes for beta-catenin, the IHC protocol described by Tandon et al. is used widely in cancer diagnostics in clinical pathology departments, which have to be held to a high standard because test results are used to subclassify cancers and guide patient care. The "staining" shown in Fig. 7C shows nonspecific DAB precipitates and are not worthy of publication in Nature Communications. Furthermore, there is no evidence presented, here or elsewhere, for a role for Notch2 in stromal cells in CLL within human tissues. Absent such connections, the significance of the phenomena described in this paper to human CLL is uncertain.

We disagree with this reviewer that IHC Figure shows nonspecific DAB staining. (All staining were performed at high standards by the Haematology and Oncology Diagnostic

Service (HODS) at Addenbrooke's hospital and analysed by Dr Livia Rásó-Barnett, who is an experienced Consultant haemato-pathologist). This staining pattern is consistent with data from Derksen et al (PNAS 2004; PMID 15067127; Figures 2B and 2C- as we discussed in our 1st submission paper).

We may reemphasise that we had taken this reviewer's criticism very seriously and provided additional evidence for beta-catenin staining in tissues using IF (Appendix figure 12 in our first revised manuscript's rebuttal letter). These data confirm our data from IHC. As these IF-staining may be more convincing to the reviewer and the reader, we have now included the IF data into the main figure and moved the IHC to the supplemental data.

Reviewers' Comments:

Reviewer #2:

Remarks to the Author:

The authors have addressed my questions and critiques adequately by providing additional data to suggest an on-target mechanism (with respect to their proposed model) of DAPT treatment in MSCs in an in vivo model.

Reviewer #3:

Remarks to the Author:

Comments for Authors

The authors have been responsive or have provided plausible explanations in response to most of my prior comments.

Original comments 1, 6, and 7: A recent report from the Elowitz lab suggests that cis activation may also occur via productive interactions within the same cell; this may be worthy of mention (<https://www.biorxiv.org/content/early/2018/05/02/313171>).

Original comment 2: The new IF showing nuclear staining for Notch2 in stromal cells is supportive of this part of the author's proposed model and strengthens the paper.

Original comment 3: No additional comments, except that the beta-catenin IHC in CLL cells that authors provide is from lymph node, not bone marrow.

Original comment 4: The lack of reliability of commercial antibodies from some vendors is a problem for the field, and those from Santa Cruz are among the chief offenders (which is not to say that all of their antibodies are unreliable; some are okay). In our experience, for example, <50% of antibodies from this vendor that are "approved" for IHC on tissues actually stand up to rigorous testing. Referring to other papers in which the antibodies are used does not address the issue. The only way to do so is to conduct in-house proof of specificity studies, e.g., CRISPR knockout of JAG2 in a line known to express JAG2 based on other measurements (e.g., RT-PCR analysis).

Original comment 9: No additional comments

Original comment 18: The authors response is unconvincing; lymph node-based CLLs have been stained with reliable IHC methods for beta-catenin by many groups, most led by hematopathologists, and in most reports most cases are negative. It may be that small amounts of beta-catenin are generated via the signaling loop proposed by the authors, and perhaps this is sufficient to have important pathophysiologic effects, but the staining shown in Fig. 7C will not impress people in the field who are familiar with IHC staining results for beta-catenin.

Point-by-point response to the Reviewer's comments

Reviewer #3

Original comments 1, 6, and 7: A recent report from the Elowitz lab suggests that cis activation may also occur via productive interactions within the same cell; this may be worthy of mention (<https://www.biorxiv.org/content/early/2018/05/02/313171>).

We thank the reviewer for this comment. Following his/ her recommendation, we are now discussing this possibility in our discussion section. The phrase in the discussion section "*Furthermore, the expression of Notch-receptors and Notch-ligands on the same cell can lead to cis-inhibition of Notch signalling. Therefore, the transmitted signal strength is also regulated by the stoichiometric relation between ligands and receptors Ref37*" has been replaced by the following text:

"In addition, CLL cells may provide conditions to MSCs permitting the recently suggested form of cis-activation of Notch signalling Ref37".

Original comment 4: The lack of reliability of commercial antibodies from some vendors is a problem for the field, and those from Santa Cruz are among the chief offenders (which is not to say that all of their antibodies are unreliable; some are okay). In our experience, for example, <50% of antibodies from this vendor that are "approved" for IHC on tissues actually stand up to rigorous testing. Referring to other papers in which the antibodies are used does not address the issue. The only way to do so is to conduct in-house proof of specificity studies, e.g., CRISPR knockout of JAG2 in a line known to express JAG2 based on other measurements (e.g., RT-PCR analysis).

We agree with this reviewer that there is a great variety in the quality of antibodies with a seeming inverse correlation between the number of offered products and quality. The suggested in-house proof is certainly a way to test how reliable this antibody is. As this would take us several weeks to months to validate, we feel that this is not justified given how little this would affect the overall conclusions. More importantly and as discussed previously, we have used a different antibody for FC (JAG1, MHJ1-152 BD Bioscience) to validate expression of JAG1 on CLL cells. In the Kamdje paper, a JAG1 Ab from Cell Signaling Technology was used. Therefore, we are reasonably comfortable that JAG1 is expressed on CLL cells. However, if the reviewer insists, we can remove the JAG1 immunoblot from the supplemental data.

Original comment 18: The authors response is unconvincing; lymph node-based CLLs have been stained with reliable IHC methods for beta-catenin by many groups, most led by hematopathologists, and in most reports most cases are negative. It may be that small amounts of beta-catenin are generated via the signaling loop proposed by the authors, and perhaps this is sufficient to have important pathophysiologic effects, but the staining shown in Fig. 7C will not impress people in the field who are familiar with IHC staining results for beta-catenin.

As discussed in our previous rebuttal letters, we agree with the reviewer that there remains controversy about the detection of beta-catenin by IHC. Our IF data however show positivity for activated beta-catenin in Ln sections. As suggested by the editor, we have now included all the cases investigated as a supplementary figure to strengthen

our conclusions. To provide a balanced discussion on this topic, we have further amended our previous statement in the discussion section: *“The degree of β -catenin activation by stroma-derived signals certainly depends on the duration of these cell-cell interactions in vivo as well as on the tissue concentrations of soluble, Wnt-regulating factors. It remains to be experimentally addressed to what extent BMSCs-induced β -catenin regulates gene-expression in tissues. For this, we currently perform CHIP-seq experiments on stroma-activated CLL cells.”*